**Measurement Report: A Multi-Year Study on the Impacts of Chinese New Year Celebrations on Air**
**Quality in Beijing, China.**
Benjamin Foreback[1,2], Lubna Dada[2], Kaspar R. Daellenbach [2], Chao Yan[1,2], Lili Wang[3], Biwu Chu[2,4], Ying
Zhou[1], Tom V. Kokkonen[2,6], Mona Kurppa[7], Rosaria E. Pileci[5], Yonghong Wang[2], Tommy Chan[2], Juha
Kangasluoma[1,2], Lin Zhuohui[1], Yishou Guo[1], Chang Li[1], Rima Baalbaki[2], Joni Kujansuu[1,2], Xiaolong Fan[1],
Zemin Feng[1], Pekka Rantala[2], Shahzad Gani[2], Federico Bianchi[1,2], Veli-Matti Kerminen[2], Tuukka Petäjä[1,2,6],
Markku Kulmala[1,2,6], Yongchun Liu[1] and Pauli Paasonen[2]
[1]Aerosol and Haze Laboratory, Beijing Advanced Innovation Center for Soft Matter Science and Engineering,
Beijing University of Chemical Technology, Beijing, China
[2]Institute for Atmospheric and Earth System Research / Physics, Faculty of Science, University of Helsinki,
Finland
[3]Institute of Atmospheric Physics, Chinese Academy of Sciences, Beijing, China
[4]State Key Joint Laboratory of Environment Simulation and Pollution Control, Research Center for Eco-
Environmental Sciences, Chinese Academy of Sciences, Beijing 100085, China
[5]U-Earth Biotech Ltd., London, United Kingdom
[6]Joint International Research Laboratory of Atmospheric and Earth System Sciences, School of Atmospheric
Sciences, Nanjing University, Nanjing, China
[7]Atmospheric Composition Research, Finnish Meteorological Institute, Helsinki, Finland
Corresponding Author: Pauli Paasonen
Correspondence to: Pauli Paasonen (pauli.paasonen@helsinki.fi)
**ABSTRACT**
This study investigates the influence of the Chinese New Year (CNY) celebrations on local air quality in Beijing
from 2013 through 2019. CNY celebrations include burning of fireworks and firecrackers, which consequently
has a significant short-term impact on local air quality. In this study, we bring together comprehensive
observations at the newly-constructed Aerosol and Haze Laboratory at Beijing University of Chemical
Technology – West Campus (BUCT-AHL) and hourly measurements from twelve Chinese government air
quality measurement stations throughout the Beijing metropolitan area. These datasets are used together to
provide a detailed analysis of air quality during the CNY over multiple years, during which the city of Beijing
prohibited the use of fireworks and firecrackers in an effort to reduce air pollution before CNY 2018. Datasets
used in this study include particulate matter mass concentrations ($PM_{2.5}$ and $PM_{10}$), trace gases ($NO_X$, $SO_2$, $O_3$,
and CO) and meteorological variables for 2013-2019, aerosol particle size distributions, and concentrations of
sulfuric acid and black carbon for 2018 and 2019. Studying the CNY over several years, which has rarely been
done in previous studies, can show trends and effects of societal and policy changes over time, and the results
can be applied to study problems and potential solutions of air pollution resulting from holiday celebrations.
Our results show that during the 2018 CNY, air pollutant concentrations peaked during the CNY night (for
example, $PM_{2.5}$ reached a peak around midnight of over 250 $\mu g/cm^3$, compared to values of less than 50 $\mu g/cm^3$
earlier in the day). The pollutants with the most notable spikes were sulfur dioxide, particulate matter, and
black carbon, which are emitted in burning of firework and firecrackers. Sulfuric acid concentration followed
the sulfur dioxide concentration and showed elevated overnight concentration. Analysis of aerosol particle
number size distribution showed direct emissions of particles with diameters around 100 nm in relation to
firework burning. During the 2019 CNY, the pollution levels were somewhat lower ($PM_{2.5}$ peaking to around
150 $\mu g/cm^3$ at CNY compared to values around 100 $\mu g/cm^3$ earlier in the day) and only minor peaks related to
firework burning were observed. During both CNYs 2018 and 2019 secondary aerosol formation in terms of
particle growth was observed. Meteorological conditions were comparable between these two years, suggesting
that CNY-related emissions were less in 2019 compared to 2018. During the 7-year study period, it appears
that there has been a general decrease in CNY-related emissions since 2016. For example, peak in $PM_{2.5}$ in
2016 was over 600 $\mu g/cm^3$, and in the years following, the peak was less each year, with a peak around 150
$\mu g/cm^3$ in 2019. This is indicative of the restrictions and public awareness of the air quality issues having a
positive effect on improving air quality during the CNY. Going into the future, long-term observations will
offer confirmation for these trends.
**1 INTRODUCTION**
Anthropogenic emissions associated with festivities, notably fireworks and firecrackers (hereafter simply
fireworks), are known for their hazardous effects, and even short-term exposure can have significant impacts
on human health (Bach et al., 2007; Chen et al., 2011; Jiang et al., 2015; Yang et al., 2014). Firework
celebrations are found to increase the concentrations of trace gases and particle concentrations (Kong et al.,
2015; Li et al., 2013). Furthermore, some studies have related these festivities to the occurrence of haze
episodes in the days following a firework event (Li et al., 2013; Feng et al., 2012).
The Chinese New Year (CNY) is a traditional annual holiday occurring in wintertime – in January or in
February as the exact date is based on the lunar cycle. Because of the adverse impacts on health, pollution from
fireworks during the CNY has gathered attention worldwide. For instance, studies including Yang et al. (2014)
in Jinan, Shi et al. (2014) in Tianjin, and Feng et al. (2012) and Zhang et al. (2010) in Shanghai have shown
that there is noticeable degradation in air quality associated with Chinese New Year celebrations in these cities.
Wang et al. (2007) has shown that firework celebrations emit significant amounts of sulfur dioxide and black
carbon. The effects of fireworks on air pollution are known for various holidays in other countries as well.
Studies in India, for example, during the country's annual Diwali Festival in the late autumn have also shown
results of high pollution from firework use (Ravindra et al., 2003; Mönkkönen et al., 2004; Barman et al., 2007;
Singh et al., 2009; Yerramesetti et al., 2013). As another example, a study by Liu et al. (1997) in Southern
California, USA showed enhanced concentrations of particulate matter and trace gas pollutants during firework
celebrations.
Because of the rising awareness of air quality problems during holiday celebrations, the government of Beijing
decided to implement a prohibition on firework burning within the 5[th] Ring Road of Beijing, in an effort to
reduce air pollution, described in a study by Liu et al. (2019). Their study reported that the prohibition resulted
in about a 40% decrease in the total number of fireworks and firecrackers sold in the city of Beijing during the
2018 CNY holiday compared to 2016. Furthermore, Liu et al. (2019) reported that observed concentrations of
air pollutants during the 2018 CNY was significantly less than that in 2016.
Therefore, an aim of this study is to confirm the conclusions of the Liu et al. (2019) study, using not only a
2016 vs. 2018 comparison, but a longer study of each year between 2013 and 2019. Furthermore, this study
offers a spatial comparison of the area where fireworks were prohibited (inside the 5[th] ring) with a region where
there was no prohibition (outside the ring). Currently, there are no previous studies that perform such a side-
by-side comparison of areas with different firework burning policies.
This manuscript provides a detailed view on how CNY celebrations have influenced air quality and atmospheric
chemistry in the Beijing metropolitan area. We start with an in-depth analysis of data from 2018 and 2019, and
then we expand with the longer 7-year dataset. Combined, these datasets provide perspective into the impacts
of the imposed restrictions on firework use in the Beijing area. The specific questions we aim to answer include:
1.) how the CNY celebrations and associated increase in precursor and aerosol emissions reflect in the
atmospheric concentrations of trace gases and particulate matter and particle number size distribution; 2.) how
these changes are connected with meteorological conditions; 3.) how the influence of CNY affects regional air
quality variation spatially over the Beijing area; and 4.) how the influence of CNY on Beijing air quality has
changed during the recent years, including the result of the firework prohibition beginning in 2018.
**2 METHODS**
The observations used in this study include measurements collected from the Beijing University of Chemical
Technology, Aerosol and Haze Laboratory (BUCT-AHL), an academic research station in Beijing China (Liu
et al., 2020); along with seven years of data from twelve measurement stations throughout the Beijing
metropolitan area, operated by the Chinese Ministry of Environmental Protection (MEP). The long-term
datasets also provide spatial context in the scale of the greater Beijing area, including a comparison of
measurements inside versus outside of the prohibition area. Here we investigated years 2013-2019. Although
data from the 2020 CNY is available, we have decided not to include it in this study because of the widespread
impacts of the COVID-19 virus that affected China during this time. Due to the unfortunate circumstance, many
Chinese citizens refrained from travel, public celebrations and time spent in public. Consequently, the 2020
CNY is not directly comparable to previous years.
This study is novel and unique in a few ways. First, it is one of only a few studies to not only show
measurements for a single CNY (or similar celebratory holidays in other countries), but it studies the holiday
over seven continuous years. This offers the ability to show trends and effects of, for example policy changes,
over time. Furthermore, this study uses data from multiple institutions, which demonstrates the value of
collaborations between different institutions when it comes to solving major global problems such as air
pollution. This study also compares the CNY inside the centre of the city to the greater Beijing area, which is
unique compared to any previous CNY (or similar holiday) air quality study that uses data at a single location.
Our insights offer value to scientists and policy makers around the world who are interested in improving air
quality during holidays that involve firework celebrations. Improving air quality, even short-term, can have a
significant positive impact on health and wellbeing of citizens.
**2.1 Measurement sites**
This study uses data collected from two sources. First, we used data from the newly constructed station near
the third ring road of Beijing (39°56'N, 116°17'E; Figure 1; Liu et al., 2020). The station, known as the Aerosol
and Haze Laboratory, is located at Beijing University of Chemical Technology West Campus, on the roof of a
five-floor building nearby to a busy highway. The station (BUCT-AHL) follows the concept of the Station for
Measuring Ecosystem-Atmosphere Relations (SMEAR) in Hyytiälä, Southern Finland (Hari and Kulmala,
2005). BUCT-AHL was built in collaboration with the Institute of Atmospheric and Earth System Research
(INAR) at the University of Helsinki as part of the effort to build a global SMEAR network (Kulmala, 2018),
with the aim to understand atmospheric chemical cocktail in megacity (Kulmala, 2015). In addition to collecting
data for in-depth air quality analysis, this joint work increases collaboration between atmospheric scientists in
China and Finland.

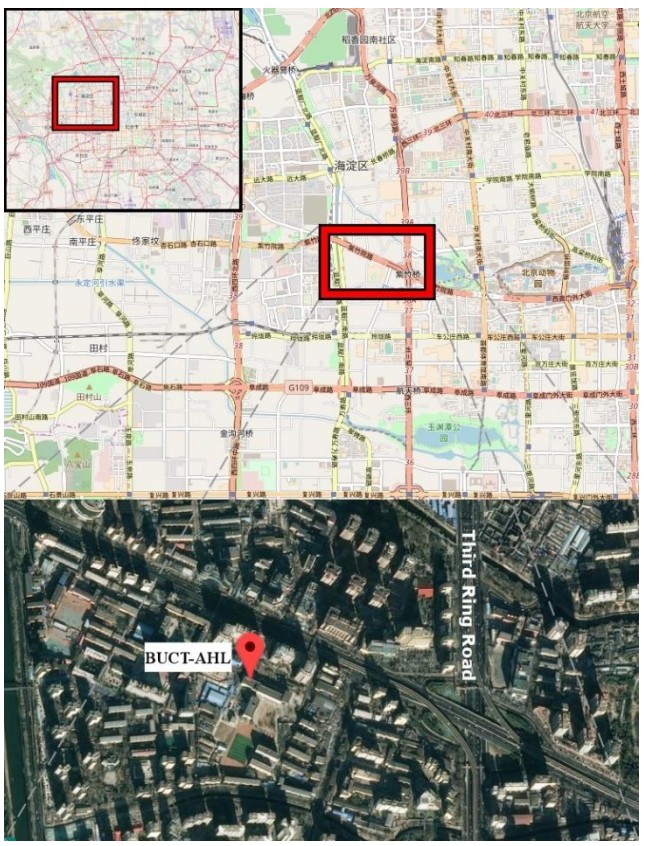

**Figure 1:** Location of the BUCT-AHL site within the Beijing metropolitan area. © OpenStreetMap
contributors YEAR. Distributed under the Open Data Commons Open Database License (ODbL)
v1.0.

In our analysis, the following datasets from BUCT-AHL during the 2018 and 2019 CNY are used: 1) Trace gas
concentrations: nitrogen oxides ($NO_X$), sulfur dioxide ($SO_2$), ozone ($O_3$), and carbon monoxide (CO); 2) Black
carbon mass concentration (BC); 3) Sub-micron aerosol particle number size distributions; 4) Gas-phase
sulfuric acid ($H_2SO_4$) concentration; 5) Meteorological observations. Technical details of the instruments,
including manufacturer, parameters measured, time resolution, and available time periods of measurements,
can be found in Table S2 in Supplementary material. These details are also described in Liu et al. (2020).

Additionally, we obtained datasets from several national air quality monitoring sites within the Beijing
metropolitan area (NAQMS; Song et al., 2017; Tao et al., 2016). These datasets were obtained from the Chinese
Ministry of Environmental Protection (MEP), which contain the following: 1) Fine and coarse particulate
matter mass concentrations ($PM_{2.5}$ and $PM_{10}$), 2) trace gases ($NO_X$, $SO_2$, $O_3$, and CO) from 2013 through 2019
for a multi-year comparison. This also provided insights into the spatial variability within the Beijing city and
particularly contrasting the area where the ban for the fireworks was implemented against the urban background
air quality.

## 2.2    Instrumentation

### 2.2.1 Observations in BUCT-AHL station

**Trace gas measurements**

Concentrations of carbon monoxide (CO), sulfur dioxide ($SO_2$), ozone ($O_3$) and nitrogen oxides ($NO_x$) were
measured with Thermo Environmental Instruments models 48i, 43i-TLE, 42i, and 49i, respectively. They were
sampled through a common inlet through the roof of the building. The length of the sampling tube was
approximately 3 m long (Zhou et al., 2020). The time resolution of the measurements was 5 minutes, but to be
consistent with the MEP datasets, one-hour averages were used in this study.

**Meteorological observations**

Meteorological datasets for 2018-2019 at BUCT-AHL were collected with a Vaisala automatic weather station,
AWS310, including wind speed and direction, ambient air temperature and relative humidity. Boundary layer
height (BLH) was measured using a Vaisala CL-51 ceilometer. Meteorological and BLH measurements were
taken on the rooftop of BUCT-AHL.
Archived meteorological data for Beijing from 2013-2017 was obtained from the Weather Underground
website (https://www.wunderground.com/history/daily/cn/beijing/ZBNY/). The station used is the Beijing
Nanyuan Airport (ICAO identifier ZBNY), a small airport located between the 4[th] and 5[th] Ring Road, south of
Beijing city center. The station is approximately 17 km from BUCT-AHL.

**Sub-micron aerosol particle number size distributions and total number concentrations**

Particle size distribution (PSD) between 3 nm and 1µm was measured using an instrument of the same name,
PSD (Liu et al., 2016). The instrument is composed of a nano-scanning mobility particle sizer (nano-SMPS, 3–
55 nm, mobility diameter), a long SMPS (25–650 nm, mobility diameter) and an aerodynamic particle sizer
(APS, 0.55–10 µm, aerodynamic diameter). It was fitted with a cyclone to remove particles larger than 10 µm
from entering the system. Sampling was done from the rooftop using a 3 m long sampling tube. Additional
information about the setup of these instruments can be found in Zhou et al. (2020).
Aerosol particle sizes have been further divided into four modes, based on particle diameter: cluster mode (sub-
3 nm), nucleation mode (3–25 nm), Aitken mode (25–100 nm), and accumulation mode (100–1000 nm). The
method is described in Zhou et al. (2020).

**Gas-phase sulfuric acid**

Sulfuric acid was measured by a chemical ionization atmospheric-pressure interface time-of-flight mass
spectrometer equipped with a nitrate chemical ionization source (CI-APi-TOF, Jokinen et al., 2012). The
ionization was done with $NO_3-$ as the reagent ion in ambient pressure (e.g., Petäjä et al., 2009). Nitrate reagent
ions were produced by ionizing a mixture of 3 mL.min$^{-1}$ ultrahigh purity nitrogen flow containing nitric acid
with 20 mL.min$^{-1}$ zero air with an X-ray source. This mixture acted as the sheath flow and was introduced into
a coaxial laminar flow reactor concentric to the sample flow. The sample flow was 8.8 L min$^{-1}$ but only 0.8
L.min$^{-1}$ was drawn into the pinhole of the TOF. The sampling line was 1.6 m long stainless-steel tube having
an inner diameter of 3/4 inch and positioned horizontally. The instrument was calibrated with known
concentrations of sulfuric acid. Further information about the calibration procedure can be found in Kürten et
al. (2012).

**Black carbon mass concentration**
An aethalometer AE33 (Magee Scientific) monitored the light absorption related to the aerosol. Equivalent
black carbon (eBC) was computed based on the change in time of the light attenuation using procedures
presented in Virkkula et al. (2015).

**2.2.2 Chinese MEP Data**

Beginning in 2013, the Chinese Ministry of Environmental Protection (MEP) began installing a China-wide
network of air quality monitoring stations to measure local and regional air quality. Real-time datasets from
this sensor network are published hourly by the China Environmental Monitoring Center (CEMC), which
includes $PM_{10}$, $PM_{2.5}$, $SO_2$, $NO_X$ and CO. There are over 1000 active sensors across China (Song et al., 2017;
Tao et al., 2016).

In this study, data from 12 MEP sites throughout Beijing are used (See Table 1 in Supplementary Information
for a list of these sites and their locations). The Guanyuan (GY) site is the closest site to BUCT-AHL, about 5
km east. The original data are available at http://106.37.208.233:20035/ and for this study we have removed
the outliers with criteria presented by Wu et al. (2018).

**2.2.3   Back-trajectories with Hysplit**

Back trajectories to the BUCT-station were calculated using Hybrid Single-Particle Lagrangian Integrated
Trajectory (Hysplit). This model is developed by National Oceanic and Atmospheric Administration (NOAA)
Air Resources Laboratory and the Australian Bureau of Meteorology Research Centre, and it is one of the most
widely-used models to determine the origin of an air mass (Stein et al., 2015). In this work, Hysplit trajectories
were calculated for the CNY each year from 2013-2019, with the trajectories arriving between 18:00 and 06:00
local time (UTC+8) during the CNY. This adds value to the analysis in two ways: First, it can show whether
the air masses in Beijing originated over other urban areas in China, e.g. the greater Beijing-Tianjin-Hebei
(BTH) area, or whether the air mass came from more rural areas, e.g. Inner Mongolia or Mongolia.
Additionally, it gives a synoptic overview of the weather conditions leading up to CNY. This in turn provides
information on whether the air mass is more stagnant within the BTH area, which would result in higher
pollution buildup, or whether it originated farther away, which would mean it would be cleaner from the start
(Wang et al., 2010; Chen et al., 2015; Zhu et al., 2020).

**3  RESULTS AND DISCUSSION**

Higher atmospheric concentrations due to elevated pollutant emissions during the Chinese New Year were
observed both at BUCT-AHL and the MEP sites during the analysis periods. The observed features include
sudden spikes in concentrations of trace gases, aerosol particles, and BC. These observations agree with the
previous studies showing a connection between holiday-related firework celebrations and degraded air quality
(Jiang et al., 2015; Yang et al., 2014; Shi et al., 2014; Feng et al., 2012; Zhang et al., 2010). In the sections
below, we will delve into these results, which can broaden scientific understanding of the impacts of firework
celebrations on local and regional air quality, especially in the context of a wide metropolitan area over the
course of several years.

## 3.1 Characteristics of air quality during the Chinese New Years 2018 and 2019


The CNY was on February 16, 2018 and February 5, 2019. Figure 2 shows a timeseries of air pollutant
concentrations from eight days before to eight days after the 2018 and 2019 CNY at BUCT-AHL (except for
$PM_{2.5}$, which is from the nearby MEP sites). We observed sharp peaks in Particulate Matter mass ($PM_{2.5}$), $SO_2$,
sulfuric acid, CO, BC, NO, and $NO_2$ and ozone during firework events. In 2018 the peak in $PM_{2.5}$ was over 250
$µg/m^3$, compared to less than 50 $µg/m^3$ half a day before, and in 2019 the peak of $PM_{2.5}$ was over 150 $µg/m^3$
compared to less than 100 $µg/m^3$ earlier in the day. Similar spikes in BC, gas phase sulfuric acid, and trace gas
concentrations of several times the values earlier in the day were observed in 2018 as well.

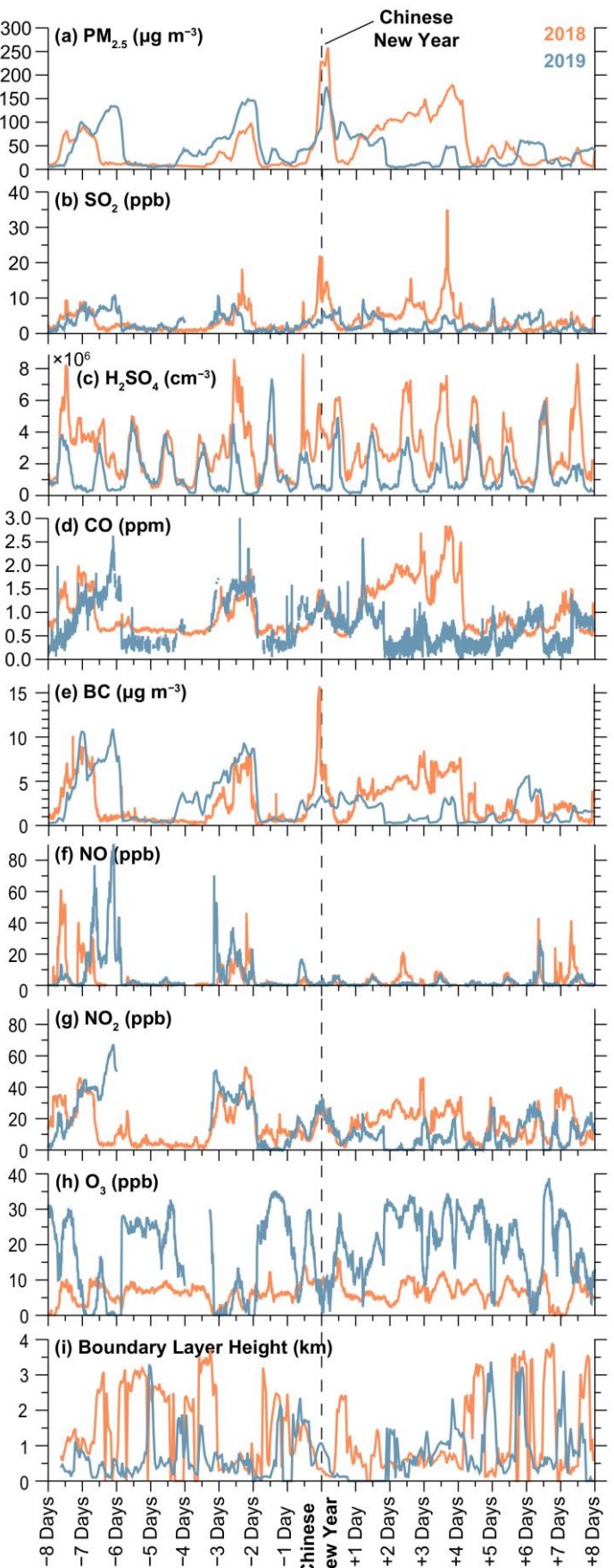


**Figure 2:** Concentrations of main pollutants measured and Boundary Layer Height in Beijing during the 2018 CNY (orange) and 2019 CNY (blue)

In contrast, in 2019, $PM_{2.5}$ was observed to have less noticeable enhancement in concentration. While there was a noticeable spike in $SO_2$ overnight of the CNY in 2018 (a spike over 20 ppb compared to less than 5 ppb earlier in the day), shown in Figure 2, a much less noticeable enhancement of $SO_2$ was observed overnight of the 2019 CNY (a peak around 5 ppb compared to around 3 ppb earlier in the day).

The measurements showed elevated nighttime concentration of $H_2SO_4$ on CNY in 2018 exceeding $3 \cdot 10^6$ cm$^{-3}$ during the whole night, which was an order of magnitude higher than typical nighttime $H_2SO_4$ concentrations of $5 \cdot 10^5$ cm$^{-3}$ (Dada et al., 2020). In 2019, there was no evident indication of anomalies in nighttime $H_2SO_4$ concentration during CNY. An unknown spike in $H_2SO_4$ was noticed at noon the day before CNY in 2018, and its association with celebratory activities is unclear. Like with $PM_{2.5}$ and $SO_2$, Figure 2 shows a distinctive spike in BC around midnight of the 2018 CNY. Although $SO_2$ and BC also originate from coal combustion and other emission sources (Wang et al., 2018), because of the shortness of the peak, and the fact that it occurs at exactly midnight, these simultaneous peaks of BC and $SO_2$ during the nighttime of CNY most likely originate from firework burning.

However, there appeared to be little to no effect of CNY on BC in 2019. The measurements showed an elevated concentration of $NO_2$ overnight of the CNY in both years (45 ppb in 2018 and 20 ppb in 2019), yet no obvious spike in NO concentration. A high $NO_2/NO_x$ ratio can be caused by accumulation of pollutants emitted the previous afternoon (Chou et al., 2009), but in case of CNY night it is straight forward to conclude it is due to firework burning, which has been shown to emit $NO_2$ but no NO (Jiang et al., 2015).

Figure 2 also shows that during the CNY celebrations in 2018 concentrations of the primary pollutants, $SO_2$, CO, BC, NO and $NO_2$, were elevated, implying enhanced direct emissions during the CNY period. Secondary pollutants are formed through chemical reactions (Seinfeld and Pandis, 2016) including, for instance, sulfuric acid and ozone. The concentrations of these secondary pollutants were as expected: sulfuric acid concentration increased due to enhanced formation rate with increased $SO_2$ concentration, and ozone concentration decreased with increased chemical sink by $NO_x$ and CO (and probably other carbon compounds). However, in 2019, only the concentrations of CO and $NO_2$ were observed to increase during CNY celebrations, leading to a decrease in ozone concentration.

Interestingly, in addition to the short-term enhancement of pollutant concentrations, Figure 2 shows degraded air quality between 16-20 February 2018, following the Chinese New Year, which closely resembles the characteristics of a haze event as described in Zhao et al. (2013), Zhao et al. (2011) and Guo et al. (2020). Using the data from BUCT-AHL, this period was quantifiably classified as a haze event using the algorithm in Zhou et al. (2020). These haze events have elevated concentrations of pollution continuously for multiple days, and concentrations gradually increase throughout the episodes. The haze eventually ends with sudden decline, often caused by an arrival of a cold front or change in synoptic weather conditions. Several previous studies, including Jiang et al. (2015) and Li et al. (2013), suggest that fireworks likely contribute to haze formation. It is plausible that the increased level of pollutants observed overnight during the 2018 CNY likely contributed to this subsequent haze period. However, the meteorological conditions and air mass origins are also important for haze formation, which are discussed in Section 3.2.

**3.2 Effects of Meteorology and Boundary Layer Height**

Because the meteorological conditions during CNY vary between different years, it is important to address the
impact of local and synoptic scale meteorological parameters on air pollution when comparing different years
to each other. Specifically, wind speed and direction, relative humidity (RH), boundary layer height, and
precipitation can affect pollutant concentrations during and after the fireworks.
However, none of the measured local meteorological variables showed drastic differences between CNY nights
of 2018 and 2019. The wind speed during the night of the 2018 CNY peaked at ~2 m/s, and during the night of
the 2019 CNY, it remained to values less than ~1 m/s (Figure 3 and Figure S1 in Supplementary). Temperature
was between 0 and 5 ºC on both years. Some difference was observed in relative humidity, as CNY 2018 took
place in very dry conditions (RH ~ 20%), whereas during CNY 2019 RH was roughly 40%. Precipitation was
not measured at BUCT-AHL in either year, and weather data measured at ZBNY show there was no
precipitation in the region during either of the years (data obtained from Weather Underground), which was
supported by observed RH values below 50%. The nocturnal boundary layer heights were less than 500 meters
in both years (Figure 2), which is unfavorable for vertical mixing of the pollutants.  Due to the slightly lower
wind speeds in 2019 than 2018, we would expect more efficient dispersion of pollutants, and thus lower
concentrations in 2018. Higher RH is also often related to higher concentrations of aerosol pollutants (Sun et
al., 2013). However, what we observed was that there were higher concentrations in 2018 than 2019. This
indicates that the reason for lower pollutant concentrations in 2019 is not due to differences in the local
meteorological conditions.

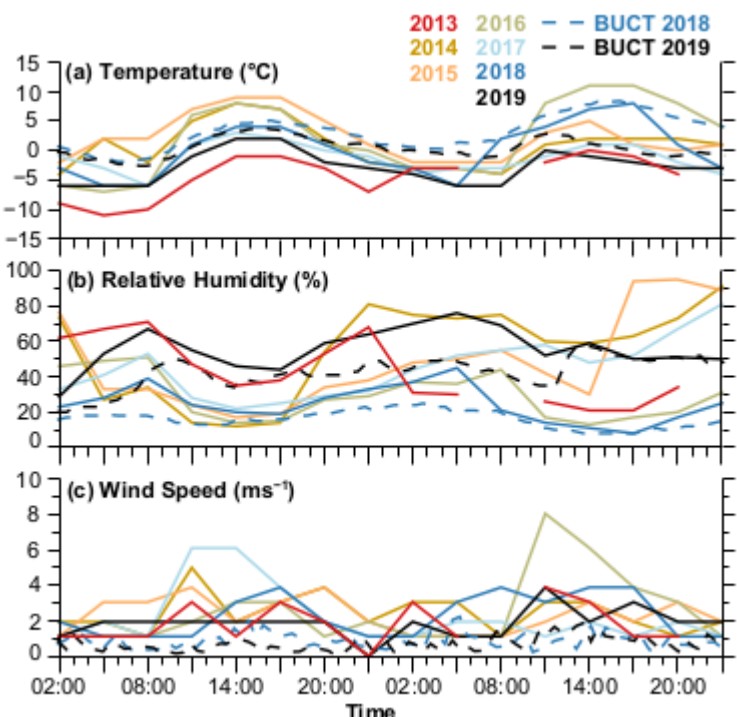

**Figure 3:** Meteorological conditions during CNY night ± one day measured in Beijing from 2013-
2019.Solid lines are measurements from Beijing Nanyuan Airport (ZBNY). These measurements are
every three hours. Dashed lines are measurements at BUCT-AHL during 2018-2019, with time
resolution of one hour.
The lower concentrations observed during the emission spike in 2019 can be either due to lower emission rates
in the area with which the measured air mass is in contact, or due to a shorter exposure to roughly similar
emissions during both years. Figure 4 shows 96 hour back-trajectories by Hysplit, during the night of CNY in
2018 and 2019, showing the sources of the airmasses. This provides further insights into the history of the
airmasses in Beijing, including how clean we can expect the airmasses to be before CNY, and whether the
airmasses are stagnant around Beijing or whether clean air is being transported into the city.
These trajectories show the following: In 2018, the airmasses from six hours prior to CNY through CNY are
from the southwest, and from two through six hours after CNY, the airmass is from the west. In 2019, airmasses
from six hours prior to CNY through two hours prior to CNY the airmasses are from the east, and following
the CNY the airmasses are primarily from the west.
Based on Wang et al. (2019), airmasses from the east are expected to be cleaner than from the southwest due
to more diffusion and less emissions from industry. However, we observed the opposite: From six through two
hours prior to midnight (i.e. the background value before the spike in pollution), the background pollutant
concentrations are higher in 2019 than in 2018. This gives further indication that the emission sources are likely
localized and short-term as opposed to long-range transport.

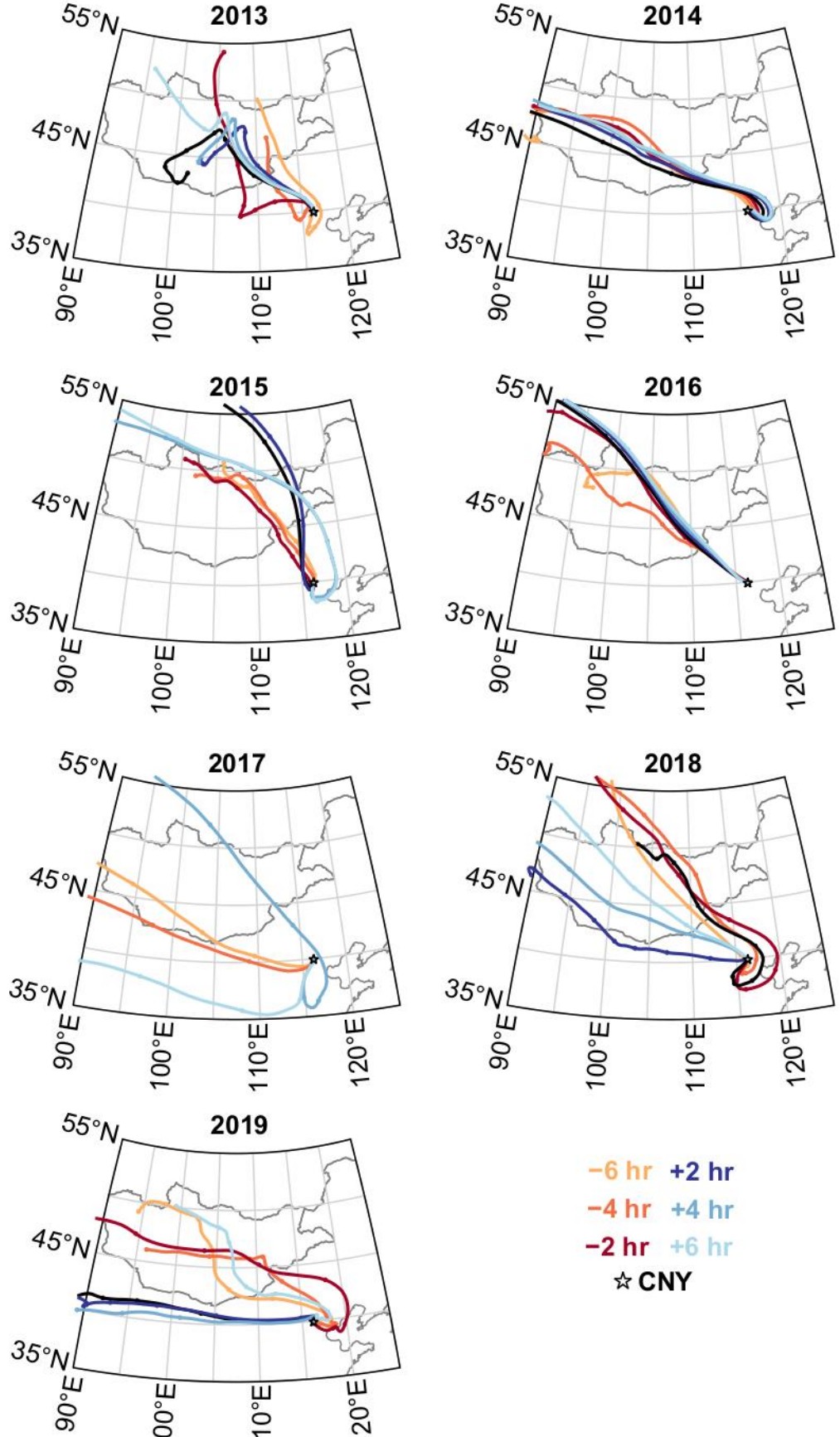

**Figure 4:** Hysplit 96 hour back-trajectories for airmasses arriving at BUCT-AHL between 18:00 and 06:00 local time, the night of CNY in 2013 through 2019. The markers are every 12 hours.

### 3.3     Aerosol particle number concentrations and aerosol number size distribution

Further exploring the effects of the fireworks on air pollution, Figure 5 shows PSD at BUCT-AHL from the day before to the day after CNY. Shortly before midnight on CNY in 2018, an elevated concentration of aerosol particles with diameters of roughly 100 nm was observed, simultaneous to the spike in $SO_2$ concentration. After the spike, $SO_2$ concentration remained elevated until the next morning. $PM_{2.5}$ concentration increases simultaneously with the $SO_2$ concentration but did not show the same spike as $SO_2$. $PM_{2.5}$ concentration remained high ($>200\,\mu g/m^3$) until the next morning, when it decreased to low values ($<30\,\mu g/m^3$) together with decreasing $SO_2$ concentration. The nocturnal pollution episode showed a very similar pattern in both $SO_2$ and $PM_{2.5}$, despite the spike in $SO_2$ occurring together with increased number concentration of roughly 100 nm particles and BC (Figure 2e). This is consistent with air pollution from firework burning. It might have originated from a source nearby, but it can also be transported as a single strong plume from further away, e.g., from outside the 5th Ring Road which was the edge of the prohibited area for firework activity. The overnight elevated concentration of $PM_{2.5}$ and $SO_2$, excluding the $SO_2$ spike, may be related to accumulated mixture of firework and other festivity related emissions, e.g., from traffic or cooking. The accumulation of $PM_{2.5}$ seems to be related to secondary aerosol formation, since the particle size distribution shows growth of particles in the dominant particle mode during the CNY night (concentration $dN/d(\log(d_P))$ over $3.3 \cdot 10^4$ between diameters 40 and 200 nm at around 8 pm and between diameters 60 and close to 300 nm at around 4 am).

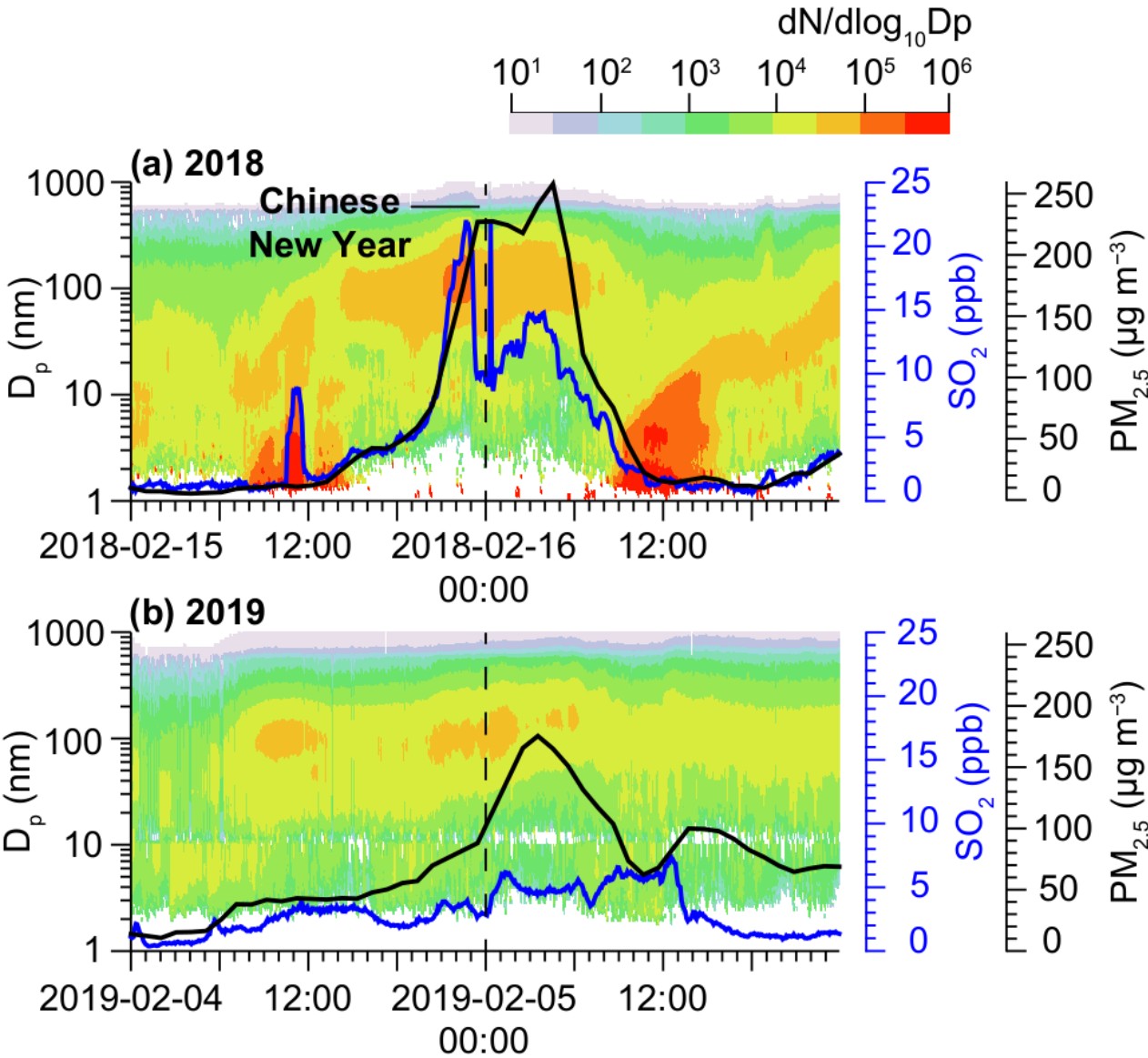

**Figure 5:** Aerosol particle number size distribution (PSD) from one day before the CNY through one day following the CNY in 2018 and 2019, overlain with aerosol mass concentration $PM_{2.5}$ (black lines) and $SO_2$ (blue lines).

In 2019, secondary aerosol mass formation was also observed, as the particle mode grew in diameter steadily between 6 pm and 6 am and the $PM_{2.5}$ concentration increased simultaneously until 4 am. The peak $PM_{2.5}$ concentration was, however, much lower in 2019 than in 2018 (roughly 100 µg/m$^3$ and close to 250 µm/m$^3$, respectively). $SO_2$ increased steadily throughout the night and exhibited only a mild peak, from 3 to 6 ppb, shortly after midnight. This peak was again accompanied with simultaneous increase in concentration of particles with diameters around 100 nm and in BC concentration (Figure 2), which suggests contribution from fireworks. However, since the $SO_2$ concentration showed only a mild peak and did not follow the $PM_{2.5}$ concentration, the contribution of nearby firework activity to the overall pollution was estimated to be negligible.

Figure 6 shows the particle number concentrations in four size modes, specifically sub-3 nm cluster mode, 3-25 nm nucleation mode, 25-100 nm Aitken mode, and 100-1000 nm accumulation mode, as a function $PM_{2.5}$

concentration measured at BUCT-AHL in 2018 and 2019. This figure starts 48 hours before CNY and runs through 48 hours after the CNY. The filled circles mark the nighttime measurements on the CNY (9pm-5am). The night-time mass concentrations are noticeably greater. The mass-to-number concentration comparison for CNY follows the same general curve during nighttime as the full time period. The pattern, particularly the nighttime observations, is consistent with recent investigation by Zhou et al. (2020), which showed that in general concentrations of pollutants are higher during nighttime, attributed to a lower boundary layer and consequent high concentrations within the boundary layer. As noted in Section 3.1, the $PM_{2.5}$ concentrations during the CNY period in 2018 were nearly an order of magnitude higher than before and after this time. The elevated PM2.5 concentration is directly connected to the elevated number concentration of accumulation mode particles (Fig. 6 bottom right panel) and the CNY data points do not diverge from the overall coupling. This indicates that the typical sizes of particles contributing to PM2.5 remains similar during CNY than before and after it. Since the accumulation mode particle concentrations form the main part of the total particle surface acting as a condensation sink for vapors forming new particles in the atmosphere and a coagulation sink for small cluster and nucleation mode particles, it is natural that the concentrations of cluster and nucleation mode decrease with increasing PM2.5 (Fig. 6, upper panels).

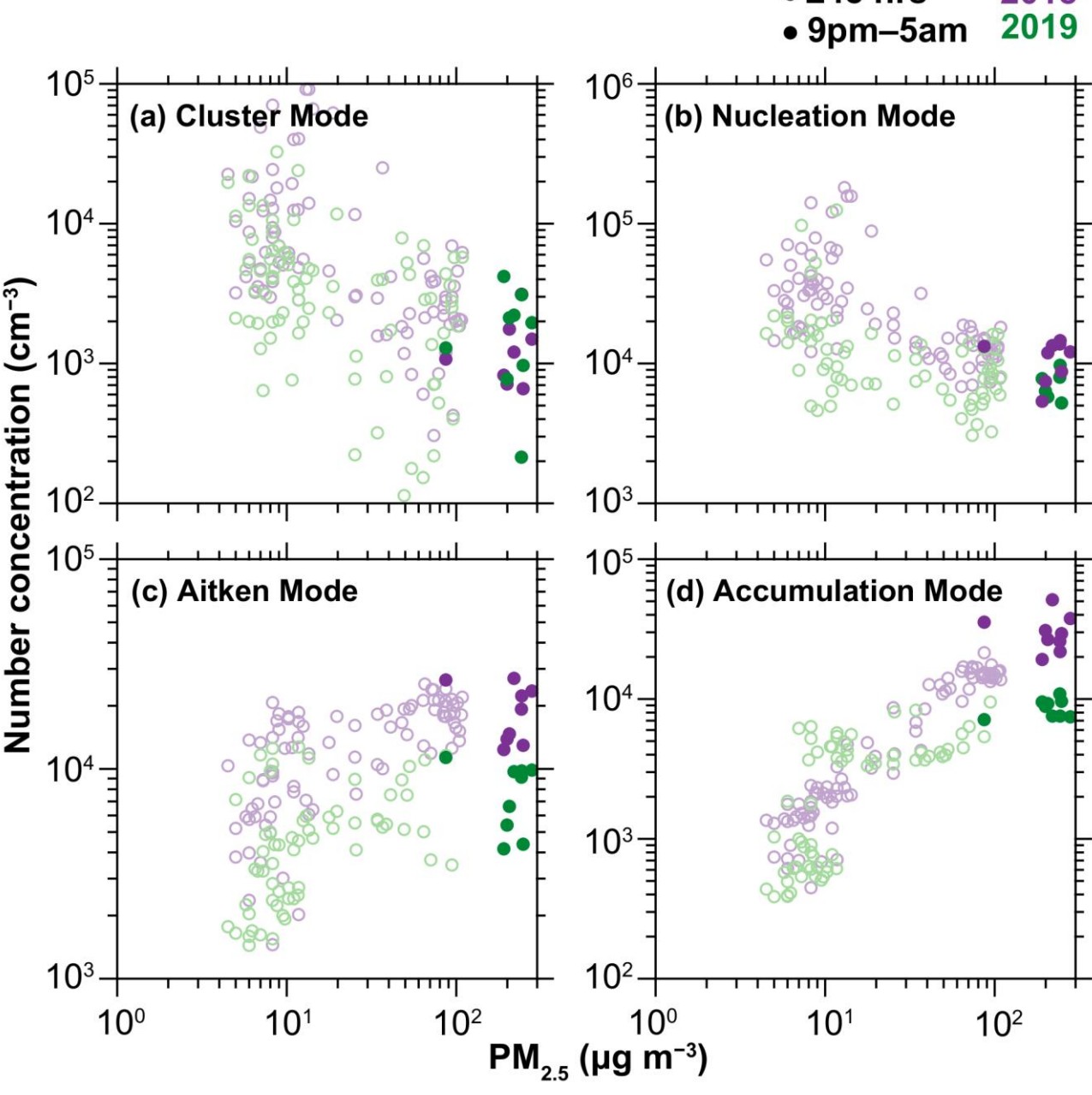

**Figure 6:** Aerosol particle number concentrations in cluster, nucleation, Aitken and accumulation modes as a function of PM2.5 mass concentration in 2018 (purple) and 2019 (green), separated from 9pm through 5am the night of the CNY (filled circles) and those of CNY ± 48 hours (open circles). The data is from BUCT-AHL.

In short, the CNY activities seem not to cause any major deviance for the typical aerosol dynamics other than the enhancement of the source of accumulation mode particles

Figure 7 depicts the cluster, nucleation, Aitken and accumulation mode particle number concentrations as a function of gas phase sulfuric acid concentration in 2018 and in 2019 inside and outside of the CNY period. Looking at the clusters, the results show a general strong dependency on the sulfuric acid as it is one of the main pre-cursors driving the process of gas-to-particle conversion (e.g, Sipilä et al. 2010, Kulmala et al. 2013, Yao et al. 2018). However, the high nocturnal sulfuric acid concentration during CNY celebrations in 2018

does not lead to high cluster or nucleation mode concentration. In fact, the particle number concentrations in
these modes deviates from the otherwise clear response to sulfuric acid concentrations. The reason for this is
visible in the panel for accumulation mode concentration vs sulfuric acid concentration: during the CNY 2018
the high concentrations of accumulation mode particles correlates with sulfuric acid concentration thus
plausibly neglecting the enhanced particle cluster and particle formation rates by enhanced coagulation sink as
explained earlier.

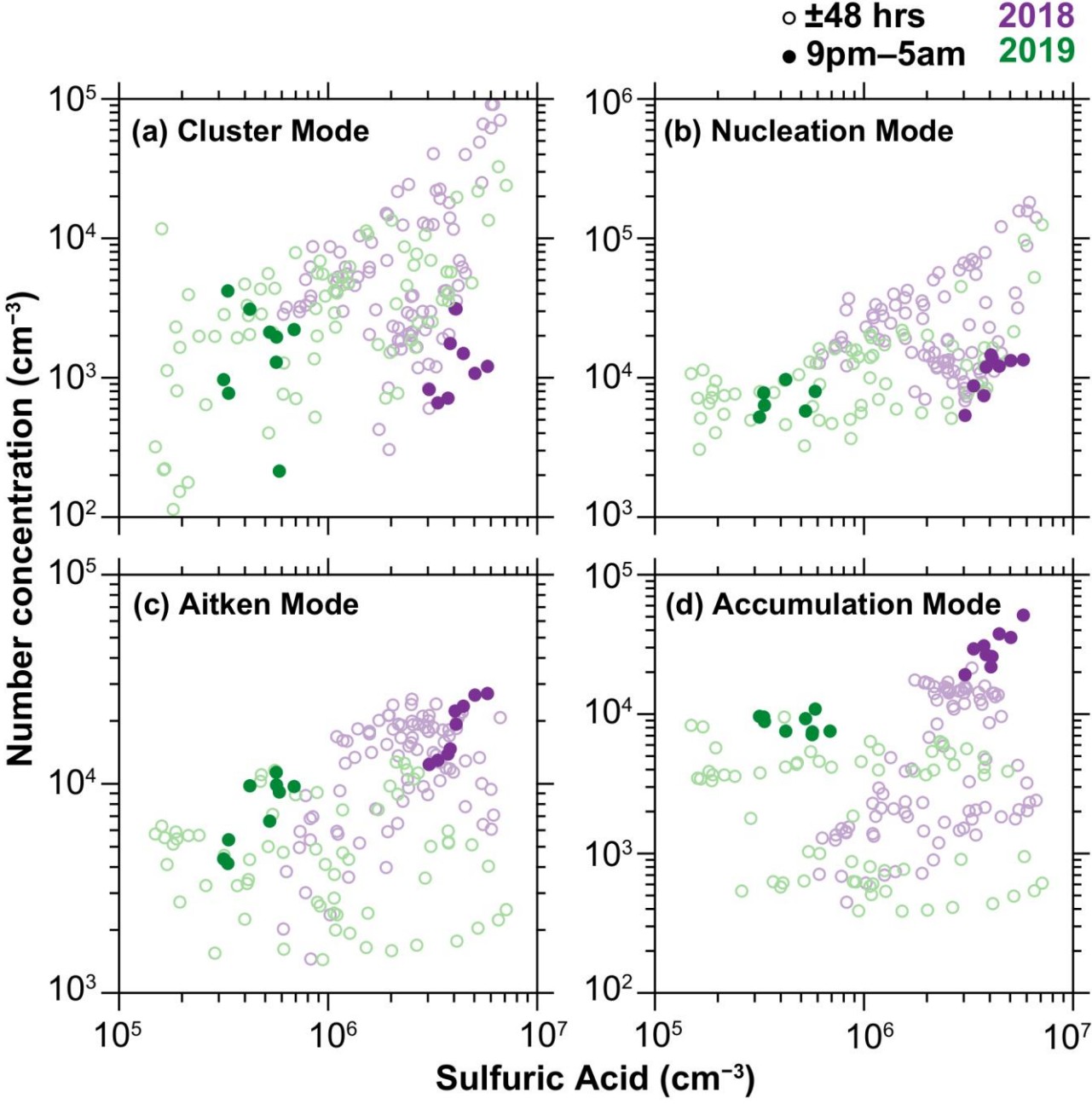

**Figure 7:** Aerosol particle number concentrations in cluster, nucleation, Aitken and accumulation modes as a
function of gas phase sulfuric acid concentration in 2018 (purple) and 2019 (green), separated from 9pm
through 5am the night of the CNY (filled circles) and those of CNY ± 48 hours (open circles). The data is from
BUCT-AHL.

**3.4 Multi-Year Variation of Chinese New Year Effects in Beijing**

Fireworks were formally prohibited within the 5[th] Ring Road of Beijing beginning in 2018, whereas outside the 5[th] Ring Road, there were no prohibitions (Liu et al., 2019). Still, there was some evidence of firework burning observed BUCT-AHL, which is within the prohibition area.

A longer-term multi-year study can be useful in demonstrating whether or not the policy is effective in reducing firework-related pollution, and if there is an overall decreasing trend of pollution effects from fireworks over multiple years. To investigate this question, it is useful to compare the 2018 and 2019 CNY with previous years in Beijing. Datasets have been analyzed from 12 MEP stations in the Beijing area from 2013 through 2019.

Figure 8 shows that each year, there was a spike in pollution around midnight during the CNY. The highest levels were observed in 2016, with the peak in $PM_{2.5}$ around midnight of the CNY reaching almost 700 µg/cm³ while values earlier in the day were less than 100 µg/cm³. The lowest levels of $PM_{2.5}$ were in 2019 with the overnight peak less than 200 µg/cm³ compared to daytime values around 50 µg/cm³. Observations from 2013, 2014, 2015, and 2017 also showed similarly high or higher levels of $PM_{2.5}$ as in 2018 (unfortunately the 2017 dataset is incomplete and does not extend beyond 00:00 of New Years day due to a network outage). The measurements for all seven years are in agreement with other studies that have linked elevated air pollution levels to CNY celebrations (Yang et al., 2014; Shi et al., 2014; Feng et al., 2012; Zhang et al., 2010), and this study further shows that the peak in 2019 is lower than in 2018, which is lower than in 2016 and 2017.

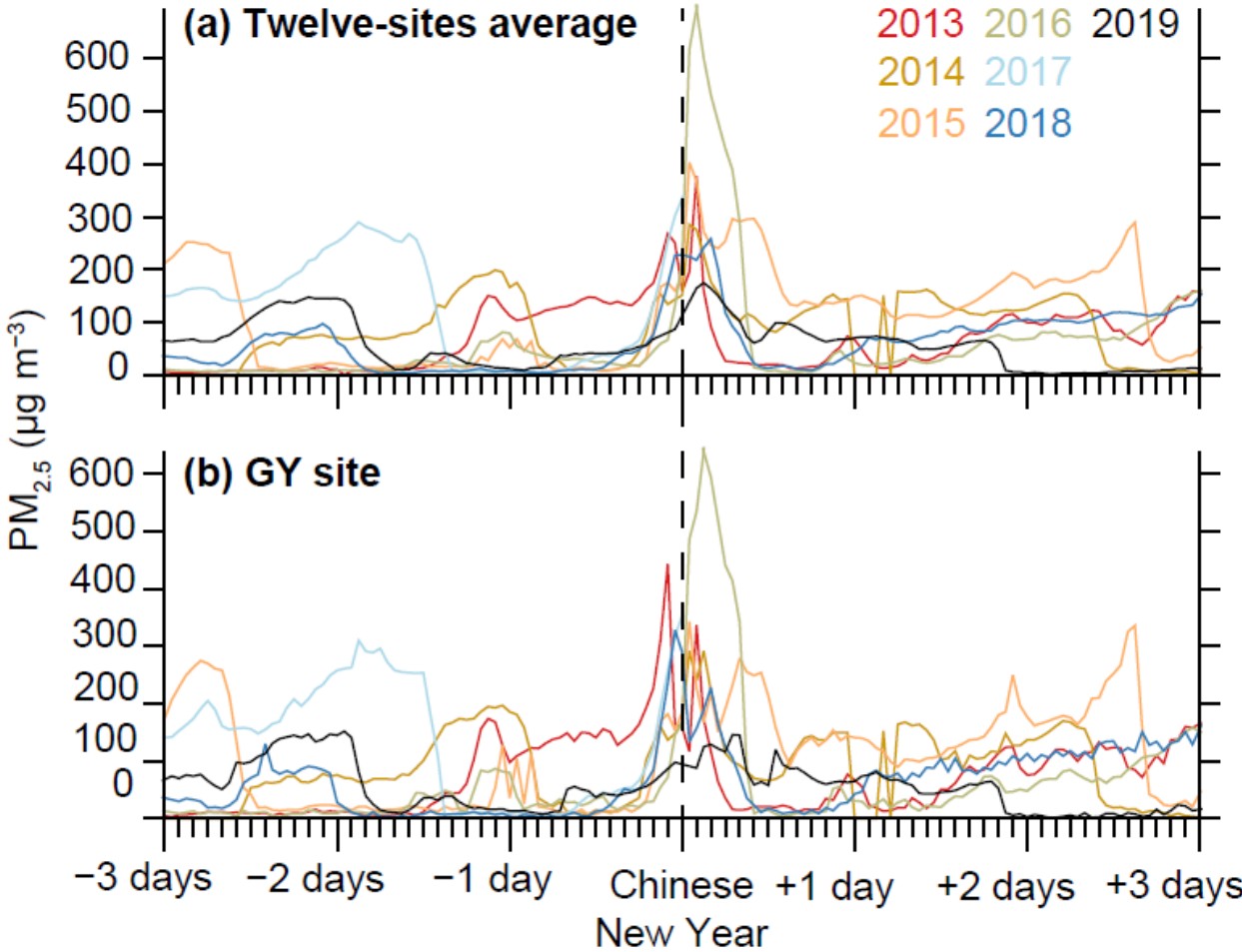

**Figure 8:** $PM_{2.5}$ averaged from 12 MEP sites in Beijing (top) and from only the Guanyuan (GY) site, which is
the closest MEP measurement site to BUCT-AHL (bottom), from three days before through three days after
the 2013-2019 CNY. The highest peak of pollution during the CNY overnight was in 2016, and the lowest was
in 2019.

Data from the CNYs have also been compiled into box plots in Figure 9, depicting the distributions of pollutant
concentrations from 6:00 pm on CNY Eve to 6:00 am on the CNY day each year at all 12 MEP stations. The
highest PM concentrations during this time were in 2016, and the $75^{th}$ and $99^{th}$ percentile concentrations have
decreased after that. On the other hand, the median concentration remained high during 2017 and 2018 but
decreased in 2019 by roughly a factor of two. Concentrations of $NO_2$ and $SO_2$ show a more steady decrease
than $PM_{2.5}$, since the median concentration of both pollutants decreased steadily from 2016 (regarding $NO_2$ for
2017), but for CO there is no clear pattern. It should be noted that in 2017, the data is missing after midnight
due to an unknown network outage. The more noticeable decrease in $NO_2$ and $SO_2$ is an expected outcome for
a ban on firework burning, since both are produced by fireworks and have shorter lifetimes than CO and $PM_{2.5}$
(Seinfeld and Pandis, 2016; Lee et al., 2011). Thus, they are less affected by long range transport and
accumulation. The decrease in pollutant concentrations since 2016 agrees with the results obtained by Liu et
al. (2019). Since ozone is a secondary product and it reacts with several primary pollutants, its concentration
pattern being roughly opposite to those of primary pollutants is as expected.

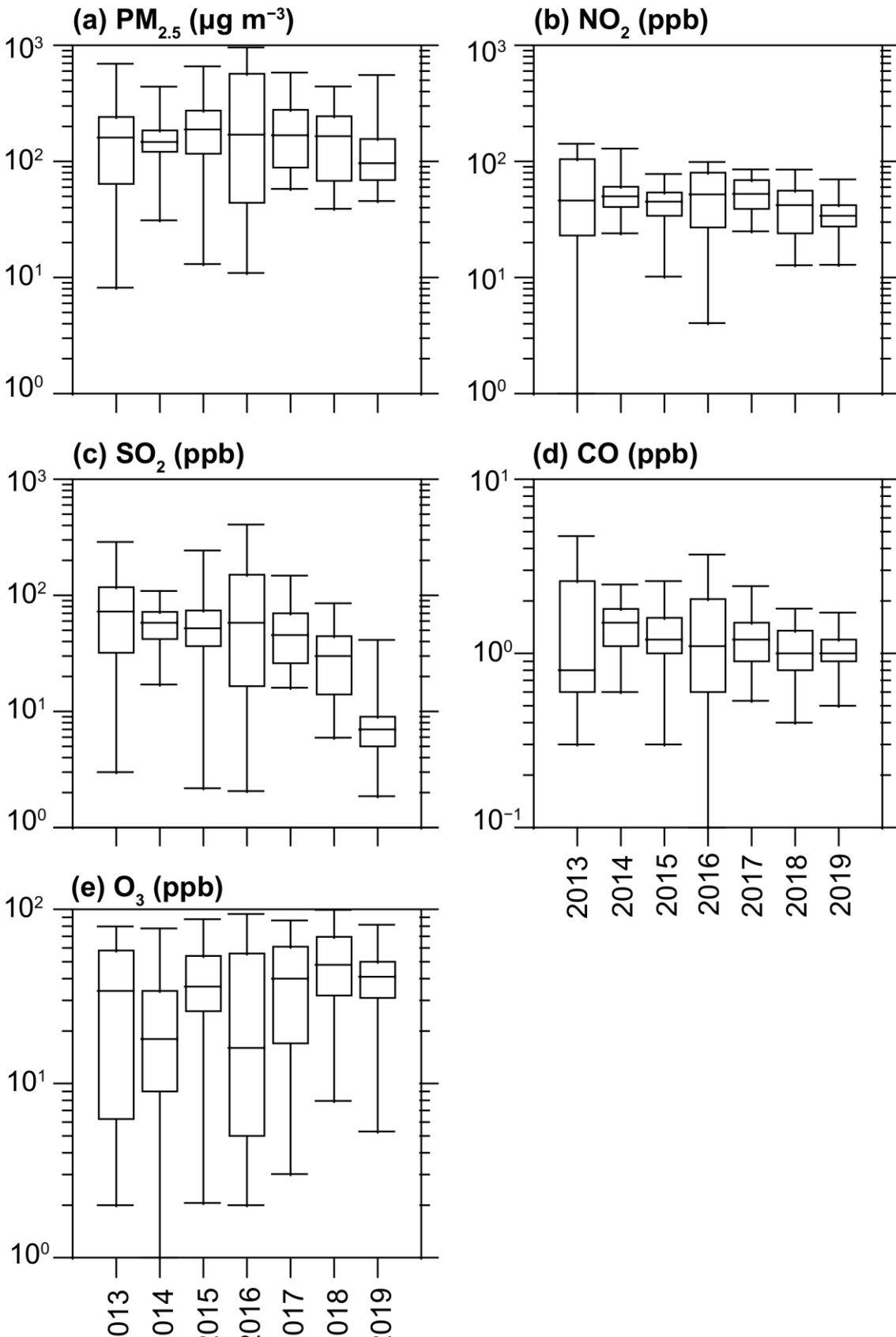

**New Years Eve (6pm–6am)**

(a) PM$_{2.5}$ (µg m$^{-3}$)

(b) NO$_2$ (ppb)

(c) SO$_2$ (ppb)

(d) CO (ppb)

(e) O$_3$ (ppb)

**Figure 9:** Boxplots of $PM_{2.5}$ and trace gases between 18:00 and 06:00 on the night of the Chinese New Year in the years 2013-2019. The boxplots show $1^{st}$, $25^{th}$, $50^{th}$, $75^{th}$, and $99^{th}$ percentiles of the data across the 12 sites during this 12-hour period (13 time points, inclusively).

Based on Hysplit back-trajectories (Figure 4), we see that in 2013-2015 the airmasses spent longer in the BTH area prior to arrival. This differs from the airmass sources in 2016-2017, where the airmasses come directly from the northwest. These areas to the northwest of Beijing, including Inner Mongolia and Mongolia usually contain less pollutants due to low anthropogenic emissions, and thus we can expect air masses from this region to be cleaner (Xu et al., 2008). Based on the airmass history, if emissions were the same, then there should be higher concentrations in 2013-2015; however, we see the highest concentrations of pollutants in 2016, followed by a decline after that. In 2018 and 2019, the airmasses spent around two days in the BTH area leading up to arrival at the station. Based on airmass source alone, we would have expected higher pollutant concentrations in 2018 and 2019, but this is not the case. Thus, we can conclude that emissions must have been highest in 2016, with lower emissions in 2018 and 2019. This agrees with Liu et al. (2019).

### 3.5    Spatial variability based on MEP measurement network data

Next, we performed a spatial comparison of the MEP measurements across the Beijing region. This includes comparing the observations inside the $5^{th}$ Ring Road, where fireworks were prohibited, to outside the ring. Figure 10 maps the 12 MEP stations in the Beijing region for 2013-2019, showing the ratio between mean $PM_{2.5}$ concentration from 9 pm through 5 am during the night of CNY and the mean concentration within $\pm$ 48 hours of the CNY at each site. Figures S2-S13 in Supplementary Information show observations of $PM_{2.5}$ from the 12 individual MEP sites and the corresponding differences, year-by-year from 2013-2019. Based on Figure 10, we can see significant variation from year to year as to which station measures the highest pollution. It is important to note that the population density is greater closer to city center, and thus the population density could impact the results. However, it is plausible to assume that the relative population density difference between the city center and the surrounding areas do not change dramatically during the few years' time period.

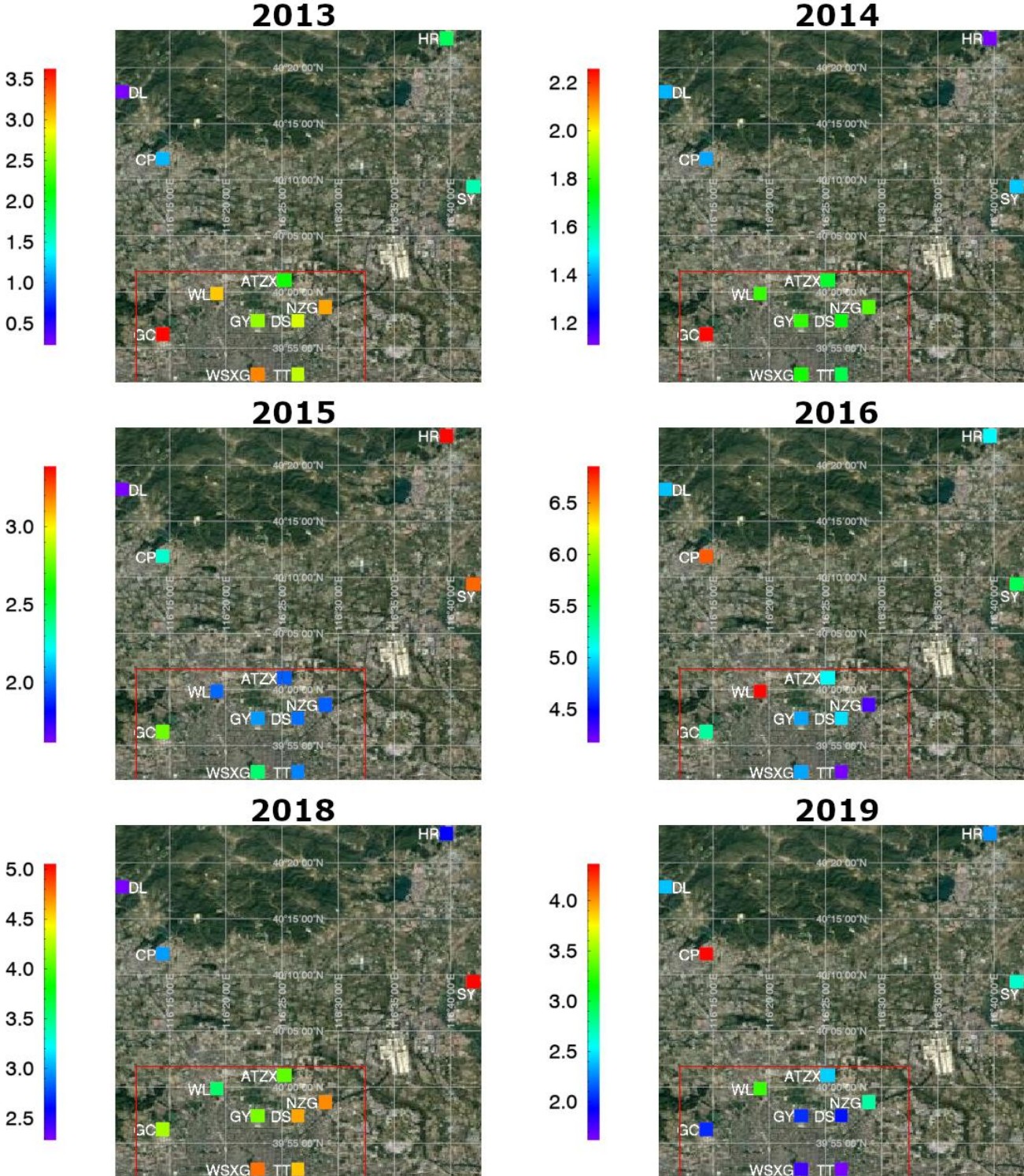

**Figure 10:** The 12 MEP sites mapped in the Beijing metropolitan area, showing the ratio of overnight PM$_{2.5}$ observations during the CNY (21:00-05:00) to all data during the period of 48 hours before through 48 hours after the CNY. The red line marks the approximate location of the 5$^{th}$ Ring Road. Note that the colorbars in each map are relative to only that year, and the colorbar range is not the same in different years. 2017 is omitted

from this figure because data after 00:00 was not available. A list of the sites' full names in English and Chinese,
along with their latitude and longitude coordinates can be found in Table S1 in Supplementary material.
Imagery: © Google Earth.
Figure 10 illustrates that in 2013 and 2014, the enhancement in PM$_{2.5}$ concentrations during CNY is greater
inside the 5$^{th}$ ring than outside. In 2015, the enhancement is much greater at the two northeastern sites (HR and
SY). In 2016, the differences vary, with no clear difference inside or outside the 5$^{th}$ ring. In 2018, the
enhancement of PM$_{2.5}$ is higher inside the 5$^{th}$ ring than outside, except for the SY site to the far northeast, which
had significantly high enhancement compared to the other sites. In 2019 the enhancement is overall less inside
compared to outside. The enhancement factors outside the 5$^{th}$ Ring road (excluding the single highest value)
and at the Northern inside stations nearest to the Ring road are quite similar in 2019, roughly in range 2.5 to 3,
but the peak times of pollution are few hours earlier at the outside stations than the Northern inside stations
(Fig. S12). The measurement sites closer to central Beijing, on the other hand, show clearly lower enhancement
factors, of values of two or below. Based on these spatial and temporal differences, and on the northerly winds
observed at the time, it is possible that the higher enhancement factors inside but close to the 5$^{th}$ Ring road are
related to emissions from outside the Ring road.
Figure 11 shows differences between the PM$_{2.5}$ mean of the sites inside the 5$^{th}$ Ring Road and the mean of the
sites outside the 5$^{th}$ Ring (that is the mean of the 8 inside sites minus the mean of the outside 4 stations) 48
hours before through 48 hours after the CNY for 2013-2019. In 2013, 2014 and 2018, the enhancement of PM$_{2.5}$
during the CNY overnight is greater inside than outside the 5$^{th}$ Ring Road. However, in 2015 and 2019, as well
as immediately after the CNY midnight in 2016, PM$_{2.5}$ was lower inside than outside. While we were lacking
the detailed data on local meteorology during 2013-2016, we were still able to analyze the meteorological
condition in terms of air mass trajectories. Figure 4 shows that, similar to 2019 as discussed previously, in 2015
and CNY midnight of 2016, airmasses arriving to Beijing were from the cleaner West-North sector and arrived
with much higher velocity in comparison to years 2013, 2014 and 2018, during which the air masses made a
turn in South or East before arrival to Beijing. Even though the CNYs during which the increase of PM2.5
enhancement inside the 5$^{th}$ Ring Road is less pronounced than outside seem to be related to faster arrival of
cleaner airmasses, we have no clear view for the reason of this difference and, due to the qualitative nature of
this comparison, it is well possible that this connection is pure coincidence. The similarity of years 2015 and
2019 in terms of the spatial variation of CNY midnight pollution peak suggests that meteorology may be at
least part of the reason for the lesser enhancement of pollution levels inside the 5$^{th}$ Ring Road than outside.
Nevertheless, the notably lower concentrations of PM$_{2.5}$ and gaseous air pollutants in 2019 than in 2015
indicates that, even with similarities in spatial distribution of changes in concentrations, the most likely reason
for lower concentrations during CNY night are the lower emissions.

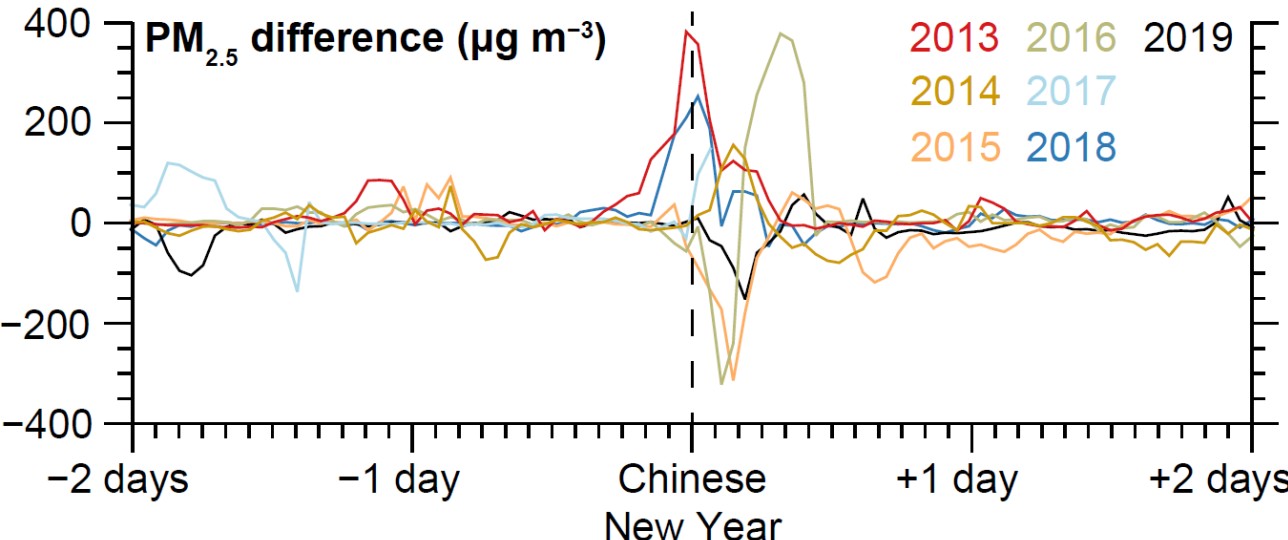

**Figure 11:** Differences between mean PM$_{2.5}$ concentrations inside and outside the 5$^{th}$ Ring Road of Beijing from 2013 through 2019. Positive values indicate higher concentration inside 5$^{th}$ Ring Road.

## 4 CONCLUSIONS

In this study, we looked at comprehensive measurements over CNY 2018 and 2019 at a measurement station in Beijing, along with long-term datasets across the Beijing metropolitan area.

Our study confirms that CNY consistently impacts air quality in Beijing. Based on our observations at the BUCT-AHL station in Beijing, in 2018, we detected higher than typical night-time concentrations of particulate mass (PM$_{2.5}$), particle number, trace gas and sulfuric acid concentrations during the CNY. This was expected, and these results are consistent with previous studies that have linked the CNY (and other similar holiday celebrations involving firework burning around the world) to degraded air quality both locally and regionally.

Our results suggest that the regulations from CNY 2018 to limit firework use have improved the air quality within the restriction zone inside the 5$^{th}$ Ring road in Beijing, and from 2016 to 2019 there has been a decrease in the effects of holiday-related pollution, which offers an optimistic outlook to the air quality impacts caused by CNY and the consequential public health concerns stemming from air pollution.

During the CNY night in 2018, we observed appearance of particles with diameters of roughly 100 nm that seemed to be linked to enhanced sulfur dioxide, sulfuric acid and black carbon concentrations, most likely as a result from firework burning. Based on the MEP data, the peaks in concentrations of different pollutants were lower than in the previous years. In 2019, a peak in pollution was observed overnight, but it was significantly lower than in 2018, while meteorological conditions were comparable in both years. The significant year-to-year variability depended presumably on the meteorological conditions. A common phenomenon for both 2018 and 2019 CNY nights was the accumulation of secondary aerosol throughout the night, seen as a diameter growth of the dominant particle mode in particle number size distributions. Measurements at BUCT-AHL showed that in 2018 a moderate haze episode began one day following the CNY, potentially related to the firework burning.

Comparing the level of increase in pollutant concentrations during CNY night inside Beijing's 5$^{th}$ ring road (firework prohibition area) to outside revealed that in 2019 the increase inside this area was smaller than outside. During most – but not all – of the previous CNYs, the increase in concentration was higher inside than

outside. This was also the case in 2018. However, as also in previous years the ratio of inside and outside concentrations during CNY has varied, it is unclear if this is related to efficacy of the emission prohibition or, e.g., to larger scale air-mass movements, or simply due to the fact that fireworks are sporadic and localized emission sources. Nonetheless, in terms of absolute concentrations, our results show a decrease of CNY pollution within the prohibition area since 2016 and especially in 2019. This is in agreement with the previous Liu et al. (2019) study, which compared the 2016 and 2018 CNY (before and after the prohibition took effect).

To conclude, this long-term analysis, which combines BUCT data with multiple years of Chinese government data at 12 locations in the Beijing area, demonstrates the importance of analyzing multiple data sources to determine overall trends, rather than making conclusions based on a single dataset. This also demonstrates the usefulness of long-term measurements. Using these datasets together, we see excellent potential that can be utilized to investigate the changes in a) atmospheric chemistry – such as ozone dynamics and sulfuric acid formation; b) atmospheric gas-to-particle conversion; c) boundary layer dynamics and d) air quality. Using CNY as a case study offers excellent insight into how rapid changes in emissions will affect air quality, health, and quality of life, especially in megacities such as Beijing. To confirm and quantify the influence of banning the firework burning in Beijing and the impact of varying meteorological conditions, similar data from coming CNYs is needed. Therefore, we suggest ongoing measurements at both BUCT-AHL and MEP sites into multiple future years.

## Acknowledgements

The work is supported by Academy of Finland via Center of Excellence in Atmospheric Sciences (project no. 272041) and European Research Council via ATM-GTP 266 (742206). This research has also received funding from Academy of Finland (project no. 316114 & 315203, 307537), Business Finland via Megasense-project, European Commission via SMart URBan Solutions for air quality, disasters and city growth, (689443), ERA-NET-Cofund as well as the Doctoral Programme in Atmospheric Sciences at the University of Helsinki. Partial support from the National Key R&D Program of China (2016YFC0200500), and the National Natural Science Foundation of China (91544231 & 41725020) is acknowledged. The authors also wish to acknowledge the Finnish Centre for Scientific Computing (CSC) – IT Center for Science, Finland, for computational resources.

## Author Contributions

All BUCT affiliated authors, plus KD, BC, YW, TC, and PR contributed to measurement collection at BUCT. LW provided the quality-controlled MEP data. BF, LD, KD, TP, FB, PP and MK conceptualized and conducted the data analysis. TK, MoK, RP, and RB participated in the data analysis. TK and MoK provided the meteorology data. KD, TP, FB, PP and MK supervised the study. BF visualized the data with assistance from SG. BF wrote the original draft and prepared the manuscript. PP, TP and all other authors reviewed and edited the manuscript.

## Competing Interests

The authors declare that there are no conflicts of interest in this study.

## Data Availability

Data from the BUCT station is available at request. Real time data from the MEP stations is available at http://106.37.208.233:20035/. Archived, quality-controlled MEP data may also be available upon request.

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
