# Peer review of "Measurement Report: A Multi-Year Study on the Impacts of Chinese New Year Celebrations on Air Quality in Beijing, China."

_Atmospheric Chemistry and Physics, 2021_

## Author Comment (AC1)

Author Comment

Reply to Referee Comments on acp-2021-192, "Measurement Report: A Multi-Year Study on the Impacts of Chinese New Year Celebrations on Air Quality in Beijing, China."

25 January 2022

Dear Editor and Reviewers:

On behalf of myself and the co-authors, I would like to thank you for your time in reviewing our manuscript, "Measurement Report: A Multi-Year Study on the Impacts of Chinese New Year Celebrations on Air Quality in Beijing, China" and for providing valuable feedback for improving this manuscript. We have taken the comments provided and have revised the manuscript.

As a summary, the most significant modification to this manuscript is that we have revised and expanded Section 3.2, "Effects of Meteorology and Boundary Layer Height" and have added a new section, 2.2, "Hysplit Back Trajectories." We have added a figure of Hysplit back-trajectories for the Chinese New Year (CNY) of years 2013-2019, which provides further details of the meteorological conditions and airmass sources leading up to the CNY. Elaborating on the meteorology provides more context of the observations and conclusions. We believe this addition addresses several of the comments from both referees and adds value to the manuscript.

Additionally, we made a fix to Figure 2, which had the time of CNY misaligned between 2018 and 2019, to Figure 6, which was using the wrong values for aerosol particle number concentrations, and to Figure 9 where the percentile values of pollutant concentrations were misrepresented. Figure 5 depicting the particle number size distributions on CNY in 2018 and 2019 has been changed from NAIS instrument to PSD instrument. This change was made because PSD shows a much wider range of particle sizes. The discussion and conclusions related to these changes have been altered and are described in more detail in the answers for the related referee questions below.

Below are the referee comments, along with our replies in blue.

**Referee 1**

Thank you for the work you have done. It is indeed interesting to note the differences in the air quality after regulations on firework/firecracker are implemented, especially with the wide suite of instrumentation you have, including the historical data from the government. Some general questions, comments, and suggestions are offered below.

There is a need to indicate what is new about this study compared to other work on CNY in the area. Are there other references that can be cited regarding studies done in the area of China in general?

We thank the referee for the appreciation shown in our work. We believe there are a few novelties in this study, in particular that it is one of the first to compare CNY over multiple years and also does a spatial

comparison over the whole metropolitan area, as opposed to only a single site. We have added the following summary (lines 111-120 in the revised manuscript):

> This study is novel and unique in a few ways. First, it is one of the only studies to not only show measurements for a single CNY (or similar celebratory holidays in other countries), but it studies the holiday over seven continuous years. This offers the ability to show trends and effects of, for example policy changes, over time. Furthermore, this study uses data from multiple institutions, which demonstrates the value of collaborations between different institutions when it comes to solving major global problems such as air pollution. This study also compares the CNY inside the centre of the city to the greater Beijing area, which is unique compared to any previous CNY (or similar holiday) air quality study that uses data at a single location. Our insights offer value to scientists and policy makers around the world who are interested in improving air quality during holidays that involve firework celebrations. Improving air quality, even short-term, can have a significant positive impact on health and wellbeing of citizens.

To answer the reviewer's second question, although there are many studies of CNY (and similar celebratory holidays in other countries) which are cited in the introduction, there are very few studies on the impacts of holiday celebrations over several years, using multiple measurement stations around a city, and that is what makes this study unique. In particular, this is the first of its kind in the Beijing area.

The objectives of the study can be further condensed and the focus of the study can be made clearer. The flow of the conclusions (which address the objectives) can also be improved with the more focused objectives.

We thank the reviewer for the suggestion, and we have revised the introductions to make the objectives more clear (lines 82-96 in the revised manuscript), and we have also revised the conclusions to more clearly address these objectives (lines 486-495 and 518-528).

The discussions can be further expounded.  Some of the discussion flow can also be improved (i.e. the transition from paragraph to paragraph can be made smoother). The discussions also need to connect the spikes in the emissions measured to the known firework sources (chemicals) in Beijing so that it can connect to what is being measured.  The link to the meteorology can also be further expounded. The discussion on the size distribution is interesting and can be enhanced. Is there something new that was observed? As for the trends, perhaps statistics can be put in (several instances of increases and decreases were mentioned with no quantification).

We thank the reviewer for the suggestion of making better connections to the air quality and meteorological conditions over the 7 years we have studied. We have taken the suggestion and have made various revisions to the flow and structure of the manuscript, and additionally, we have revised the meteorology section of results, including the addition of new figures, to offer more discussion and clarity. In terms of quantification and trends, we have now quantified the changes in concentrations in the text. However, we have not calculated any statistics for the trends, since the possible decreasing trend includes in practice only the years 2016, 2018 and 2019 (since data for CNY 2017 is incomplete), and a notable change in concentrations is observed only between 2018 and 2019. We mention this more clearly in the manuscript (Sections 3.1 – 3.3).

In the conclusions it is good to note the impacts of the CNY to air quality in terms of health standards, background air quality and other points of comparison that may be helpful to assess the impacts more objectively. Also how this study is new compared to other work can be discussed in the conclusions as well.

It is a good point to note the health impacts and consequences of CNY-related air pollution, and we appreciate the reviewer's suggestion. We have taken this suggestion and have revised the introduction to more clearly state the motivation of this study (lines 82-96), and we have expanded on the novelty and uniqueness of the study. We have revised the conclusions to more directly reflect on and address these points discussed in the introduction (lines 486-495 and 518-528).

Details are listed below:

Line 26: What aspect of the celebrations? The fireworks and firecrackers? Or including traffic?

We have added the following sentence after this line to clarify that the firework and firecracker events are the primary cause of elevated air pollution levels as follows (lines 24-25 in the revised manuscript):

> CNY celebrations include burning of fireworks and firecrackers, which consequently has a significant short-term impact on local air quality.

Line 27, 30, 78: "comprehensive analysis and detailed analysis" needs to be more specific (i.e. gases and particulates, elements, size-speciated?)

We have added the following to the abstract (lines 30-33 in the revised manuscript):

> Datasets used in this study include particulate matter mass concentrations ($PM_{2.5}$ and $PM_{10}$), trace gases ($NO_X$, $SO_2$, $O_3$, and CO) and meteorological variables for 2013-2019, aerosol particle size distributions, and concentrations of sulfuric acid and black carbon for 2018 and 2019.

Line 28: The type of "data" should be noted (how many sites/locations etc…)

We have specified that there are 12 Chinese government air quality measurement stations throughout the Beijing metropolitan area.

Line 30 and 63: …before CNY is vague, be more specific when this happened… also is this a total prohibition of the use of fireworks and firecrackers? It is not clear. What's the significance of the 5th ring road of Beijing?

The government's decision was to ban firework burning within the 5th ring road of Beijing. From Liu et al., 2019, which is cited: *"in 2018, a "prohibition" policy for [firework/firecracker burning] was applied within the urban area of Beijing (within the Fifth-Ring Road)."* We cannot speak for why specifically the government chose the 5th ring road as its definition for where fireworks/firecrackers were prohibited. Therefore, the significance of using the 5th ring in this study is to compare where fireworks/firecrackers were prohibited (inside this ring road) with where they weren't (outside this ring road). We have added the following text after this paragraph (lines 82-86 in the revised manuscript):

> Therefore, an aim of this study is to confirm the conclusions of Liu et al. (2019) study using not only a 2016 vs. 2018 comparison, but a longer study of each year between 2013 and 2019. Furthermore, this study offers a spatial comparison of the area where fireworks were prohibited (inside the 5th ring) with a region where there was no prohibition (outside the ring). Currently, there are no previous studies that perform such a side-by-side comparison of areas with different firework burning policies.

We will address the following comments at once, since they are together in the abstract:

Line 32: "significant peak… though not as strong" are not clear, can numbers be used?

Line 34: Is it possible to quantify this "decrease"?

Line 35-36: Can you note why SO2 and BC are the highest?

To address these comments, we have revised the abstract as follows (lines 36-49 in the revised version):

> Our results show that during the 2018 CNY, air pollutant concentrations peaked during the CNY night (for example, $PM_{2.5}$ reached a peak around midnight of over 250 µg/cm$^3$, compared to values of less than 50 µg/cm$^3$ earlier in the day). The pollutants with the most notable spikes were sulfur dioxide, particulate matter, and black carbon, which are emitted in burning of firework and firecrackers. Sulfuric acid concentration followed the sulfur dioxide concentration and showed elevated overnight concentration. Analysis of aerosol particle number size distribution showed direct emissions of particles with diameters around 100 nm in relation to firework burning. During the 2019 CNY, the pollution levels were somewhat lower ($PM_{2.5}$ peaking to around 150 µg/cm$^3$ at CNY compared to values around 100 µg/cm$^3$ earlier in the day) and only minor peaks related to firework burning were observed. During both CNYs 2018 and 2019 secondary aerosol formation in terms of particle growth was observed. Meteorological conditions were comparable between these two years, suggesting that CNY-related emissions were lower in 2019 compared 2018. During the 7-year study period, it appears that there has been a general decrease in CNY-related emissions since 2016. For example, peak in $PM_{2.5}$ in 2016 was over 600 µg/cm$^3$, and in the years following, the peak was less each year, with a peak around 150 µg/cm$^3$ in 2019.

Line 40: Is this improvement just for the two years? It's not clear.

We have rewritten and expanded on this for clarity. It now states (lines 49-51 in the revised manuscript):

> This is indicative of the restrictions and public awareness of the air quality issues having a positive effect on improving air quality during the CNY. Going into the future, long-term observations will offer confirmation for these trends

Line 66: Which "toxic pollutant" is being referred to?

This refers to the measurements of air pollutants. We have rewritten this section for clarity, and it now reads as follows (lines 75-80 in the revised manuscript):

> Because of the rising awareness of air quality problems during holiday celebrations, the government of Beijing decided to implement a prohibition on firework burning within the 5th Ring Road of Beijing, in an effort to reduce air pollution, described in a study by Liu et al. (2019). Their study reported that the prohibition resulted in about a 40% decrease in the total number of fireworks and firecrackers sold in the city of Beijing during the 2018 CNY holiday compared to 2016. Furthermore, Liu et al. (2019) reported that observed concentrations of air pollutants during the 2018 CNY was significantly less than that in 2016.

Line 68, 105: Maybe consider including a table of the observations made by the different instruments, including parameter measured (size range if particles) and sampling period, time resolution, and location?

We have added a table in supplementary material, Table S2, with details of the instruments used in this study. We have added the following text (lines 138-140 in the revised manuscript):

> Technical details of the instruments, including manufacturer, parameters measured, time resolution, and available time periods of measurements, can be found in Table S2 in Supplementary material. These details are also described in Liu et al. (2020).

Line 129: Indicate the time resolution of the meteorological data and other subsequent data.

We have added a table in supplementary material, Table S2, with details of the instruments used in this study, which includes the meteorological instruments.

Line 134: The first sentence needs to be improved. The PSD is measured by the PSD?

Yes, the instrument is of the same name. We have clarified and added more details about the instrument. Additionally, we have removed the NAIS and have based our analysis on this instrument instead.

Line 152: Define PSM

We added the definition of the acronym for Particle Size Magnifier.

Line 159: NO3- should be NO3- (apply the proper subscripts and superscripts for chemical names) throughout the document… in other

We have addressed this comment and have applied the subscripts to the chemical compounds, as well as to $PM_{2.5}$ and $PM_{10}$.

Line 193: Where are these "previous studies" located?  This information should actually be in the introduction. Is it possible to have a rough estimate of the increase during this time? This is to confirm the word "sharp" used to describe the increase.
Line 197: What does "significant" mean?

The previous studies are described in lines 55-73 in the introduction. We have added to this paragraph a statement quantifying the significant peak in concentrations at CNY, and we have removed the reference to previous studies here, as this had already been mentioned before and is also mentioned a few paragraphs later. This paragraph has been revised as follows (lines 242-248 in the revised manuscript):

> Figure 2 shows a timeseries of air pollutant concentrations from eight days before to eight days after the 2018 and 2019 CNY at BUCT-AHL (except for $PM_{2.5}$, which is from the nearby MEP site). We observed sharp peaks in Particulate Matter mass ($PM_{2.5}$), $SO_2$, sulfuric acid, CO, BC, NO, and $NO_2$ and ozone during firework events. In 2018 the peak in $PM_{2.5}$ was over 250 µg/m$^3$, compared to less than 50 µg/m$^3$ half a day before, and in 2019 the peak of $PM_{2.5}$ was over 150 µg/m$^3$ compared to less than 100 µg/m$^3$ earlier in the day. Similar spikes in BC, gas phase sulfuric acid, and trace gas concentrations of several times the values earlier in the day were observed in 2018 as well.

Line 198: Haze report, is there a reference for that?

We have clarified our definition of haze with the reference to Zhou et al. (2013), Zhou et al. (2011), Guo et al. (2020) and Zhou et al. (2020). This paragraph has been revised as follows, and has been moved to the end of Section 3.1 (lines 278-288 in the revised version):

> Interestingly, in addition to the short-term enhancement of pollutant concentrations, Figure 2 shows degraded air quality between 16-20 February 2018, following the Chinese New Year, which closely resembles the characteristics of a haze event as described in Zhao et al. (2013), Zhao et al. (2011) and Guo et al. (2020). Using the data from BUCT-AHL, this period was quantifiably classified as a haze event using the algorithm in Zhou et al. (2020). These haze events have elevated concentrations of pollution continuously for multiple days, and concentrations gradually increase throughout the episodes. The haze eventually ends with sudden

decline, often caused by an arrival of a cold front or change in synoptic weather conditions. Several previous studies, including Jiang et al. (2015) and Li et al. (2013), suggest that fireworks likely contribute to haze formation. It is plausible that the increased level of pollutants observed overnight during the 2018 CNY likely contributed to this subsequent haze period. However, the meteorological conditions and air mass origins are also important for haze formation, which are discussed in Section 3.2.

Line 200: "There is also a noticeable spike in SO2 ," this can be quantified.

We have added a quantification (lines 250-253 in the revised manuscript):

In contrast, in 2019, $PM_{2.5}$ was observed to have less noticeable enhancement in concentration. While there was a noticeable spike in $SO_2$ overnight of the CNY in 2018 (a spike over 20 ppb compared to less than 5 ppb earlier in the day), shown in Figure 2, a much less noticeable enhancement of $SO_2$ was observed overnight of the 2019 CNY (a peak around 5 ppb compared to around 3 ppb earlier in the day).

Line 204: What are typical background values?

We have added a reference with typical nighttime values, which provides context for the 2018 overnight measurements of sulfuric acid. The text now reads as follows (lines 255-257 in the revised manuscript):

The measurements showed elevated nighttime concentration of $H_2SO_4$ on CNY in 2018, concentration exceeding $3 \cdot 10^6$ $cm^{-3}$ during the whole night, which was an order of magnitude higher than typical nighttime $H_2SO_4$ concentrations of $5 \cdot 10^5$ $cm^{-3}$ (Dada et al., 2020).

Line 211: The discussion jumps, "however, a high NO2/NOx ratio." What did you observe of this ratio for your measurements?

We have added quantifications of the $NO_2$ peak. We chose not to show a figure of the $NO_2/NO$ ratio as it would be superfluous and wouldn't add additional value beyond what the current figure shows. The fact that there is a spike in the $NO_2$ but not NO is a signature of fireworks, which is described in Jiang et al., 2015. We have added further description to this paragraph for clarity. The revision is as follows (lines 263-267 in the revised version):

The measurements showed an elevated concentration of $NO_2$ overnight of the CNY in both years (45 ppb in 2018 and 20 ppb in 2019), yet no obvious spike in NO concentration. A high $NO_2/NO_x$ ratio can be caused by accumulation of pollutants emitted the previous afternoon (Chou et al., 2009), but in case of CNY night it is straight forward to conclude it is due to firework burning, which has been shown to emit $NO_2$ but no NO (Jiang et al., 2015).

Line 216: Should you indicate which are the primary pollutants?... sulfuric acid and ozone react to elevated concentrations of what?

We have added clarification as to which are primary and secondary pollutants. This section now reads as follows (lines 269-276 in the revised version):

Figure 2 also shows that during the CNY celebrations in 2018 concentrations of the primary pollutants, $SO_2$, CO, BC, NO and $NO_2$, were elevated, implying enhanced direct emissions during the CNY period. Secondary pollutants are formed through chemical reactions (Seinfeld and Pandis, 2016) including, for instance, sulfuric acid and ozone. The concentrations of these secondary pollutants were as expected: sulfuric acid concentration increased due to enhanced

formation rate with increased $SO_2$ concentration, and ozone concentration decreased with increased chemical sink by $NO_x$ and CO (and probably other carbon compounds). However, in 2019, only the concentrations of CO and $NO_2$ were observed to increase during CNY celebrations, leading to a decrease in ozone concentration.

Line 222,225: all the other pollutants: best to be clear which pollutants you are talking about.

We have deleted the second half of this sentence, as it was not directly relevant to the subject anyway.

Line 227: What is the link to the meteorology for your case, the past studies can be put in the introduction so you can focus on the results here.

We have expanded and elaborated about the meteorology in the hours before, during and after the CNY. We have added two new figures: Figure 3, which shows meteorology for 2013-2019, and Figure 4, which shows back-trajectories of airmasses using the Hysplit trajectory model, in each year from 2013 to 2019. This offers more insight into the ambient conditions that existed each year that would have contributed to the effects of firework emissions during CNY celebrations. This tells us, for example, how clean we can expect the airmasses to be before the firework emissions, and how quickly we can expect the air to be replaced with clean air again.

This modification also answers a similar comment by the other referee.

Line 243: How low? Quantify?

We have changed "low" to "less than 500 meters".

Line 244: This sentence is confusing, can improve.

We have rewritten this paragraph for clarity as follows (lines 305-310 in the revised version):

Due to the slightly lower wind speeds in 2019 than 2018, we would expect more efficient dispersion of pollutants, and thus lower concentrations in 2018. Higher RH is also often related to higher concentrations of aerosol pollutants (Sun et al., 2013). However, what we observed was that there were higher concentrations in 2018 than 2019. This indicates that the reason for lower pollutant concentrations in 2019 is not due to differences in the local meteorological conditions.

Line 259: Do you mean primary emissions are not observed in 2019, it's not clear in the sentence?

We have removed the first half of this sentence, and now the sentence is clearer.

Line 269: The sentence is too long, can be broken down. The end of the sentence (for 48 hours before through 48 hours after the CNY) is not clear.

We have broken this into two sentences as follows (lines 357-360 in the revised version):

Figure 6 shows the particle number concentrations in four size modes, specifically sub-3 nm cluster mode, 3-25 nm nucleation mode, 25-100 nm Aitken mode, and 100-1000 nm accumulation mode, as a function $PM_{2.5}$ concentration measured at BUCT-AHL in 2018 and 2019. This figure starts 48 hours before CNY and runs through 48 hours after the CNY.

Line 272: Did you mean the filled in circles? Rather than the darker colors?

Yes. We have changed "darker colors" to "filled circles."

Line 275: What are these recent results?

We have rewritten this sentence as two sentences as follows, which we hope addresses this comment (lines 3621-365 in the revised manuscript):

> The mass-to-number concentration comparison for CNY follows the same general curve during nighttime as the full time period. The pattern, particularly the nighttime observations, is consistent with recent investigation by Zhou et al. (2020), which showed that in general concentrations of pollutants are higher during nighttime, attributed to a lower boundary layer and consequent high concentrations within the boundary layer.

Line 276-279: There notes on lower and higher, but it is not clear in comparison to what. The discussion can be improved with references, perhaps discussed in the introduction on what are typical sizes of the firework/firecracker emissions.

We have revised this paragraph to be clearer, have added some quantification, and have added the reference. The revision is as follows (lines 365-373 in the revised manuscript):

> $PM_{2.5}$ concentrations during the CNY period in 2018 were nearly an order of magnitude higher than before and after this time. The elevated PM2.5 concentration is directly connected to the elevated number concentration of accumulation mode particles (Fig. 6 bottom right panel) and the CNY data points do not diverge from the overall coupling. This indicates that the typical sizes of particles contributing to PM2.5 remains similar during CNY than before and after it. Since the accumulation mode particle concentrations form the main part of the total particle surface acting as a condensation sink for vapors forming new particles in the atmosphere and a coagulation sink for small cluster and nucleation mode particles, it is natural that the concentrations of cluster and nucleation mode decrease with increasing PM2.5 (Fig. 6, upper panels).

Line 291: Is this area the only area with prohibitions? How far away is the next area where there are no prohibitions?

We have revised this paragraph for clarity. It is now written as follows (lines 392-394 in the revised manuscript):

> Fireworks were formally prohibited within the 5th Ring Road of Beijing beginning in 2018, whereas outside the 5th Ring Road, there were no prohibitions (Liu et al., 2019). Still, there was evidence of firework burning in the vicinity of BUCT-AHL, which is within the prohibition area.

Line 306: How is the haze apparent from the plot?

We have removed from this paragraph reference to haze episodes. We did, however, elaborate in Section 3.1 about the haze episode, and we noted that it follows the definition of a haze episode defined in Zhao et al. (2013), Zhao et al. (2011) Guo et al. (2020) and Zhao et al. (2020).

**Referee 2**

This manuscript presents a detailed analysis of air pollutant during the Chinese New Year (CNY) over multiple years in Beijing based on comprehensive datasets. The influence of CNY celebrations on local air quality is investigated. The measurements and data have been made carefully and then the interesting results are presented, especially its unique difference inside and outside the 5th Ring Road during CNY.

However, it could be more thorough when interpreting some results. In addition, there are also some language issues and editing needs that have to be addressed. The authors need to make a careful revision on the discussions to improve the overall quality of the paper for publication in ACP. I would recommend the editor to reconsider the paper after a major revision by the authors.

We thank this reviewer for the positive and helpful feedback on our manuscript. Several comments from both referees overlap with each other, and we hope that we have been able to thoroughly address the comments from both referees together.

Specific Comments:

The introduction should include the new perspectives of your work. When cite the previous studies, it is best to expound what are the difference between your work and others, rather than just laying out the results of previous research.

We thank the reviewer for the suggestion on improving the discussion on motivation behind this study, and why this study is novel compared to previous studies. We appreciate the feedback on how to improve the introduction of this manuscript.

This comment is very similar to the first comment made by Referee 1, and we believe that our revision to the introduction has addressed the comment made by both referees that the novelty and uniqueness of the study should be more clearly outlined.

Line 83: The specific question you aim to answer: (ii) how are these changes connected with meteorological conditions, the current analysis is too brief, the authors may consider adding more materials to enrich the discussion, such as the weather process before and after CNY.

We have expanded and elaborated about the meteorology in the hours before, during and after the CNY. We have added two new figures: Figure 3, which shows meteorology for 2013-2019, and Figure 4, which shows back-trajectories of airmasses using the Hysplit trajectory model, in each year from 2013 to 2019. These figures are discussed in Sections 3.2 (years 2018-2019 related to BUCT observations) and 3.4 (related to MEP observations 2013-2019). The outcome of these analyses is that the meteorological conditions alone cannot explain the decreased pollutant concentrations in 2019 and indicate that lower emissions are most likely the main reason behind the observed change. However, with the short period of data available after the prohibition of firework burning within the 5$^{th}$ Ring road, we cannot conclude whether meteorological conditions did have an additional influence on the decreased concentrations.

This modification also answers a similar comment by the other referee.

Lines 334-346: Since Fireworks were formally prohibited within the 5th Ring Road of Beijing beginning in 2018, why the enhancement of PM5 concentrations during CNY varied significantly from year to year. Please further elaborate to support your analysis. In addition, can the author give the differences of other pollutants which are link to fireworks emissions inside and outside the 5th Ring Road from 2013 to 2019?

This question by the referee is very good and it ties strongly to many of the further questions. In order to investigate the enhancement of PM$_{2.5}$ inside and outside the prohibition area, we investigated in more detail the meteorology (as the referees suggest), both in terms of local observations in 2018-2019 and of air mass trajectories for the CNY's in 2013-2019. We found that in each of the years when the enhancement of PM$_{2.5}$ was larger outside than inside the prohibition area, the air masses during the CNY midnight arrived relatively fast from the clean sector (West-North), whereas during all other years the airmasses came from other directions with smaller velocities. However, we cannot tie this connection to

any sound explanation, and thus in the revised version of the manuscript we both explain the connection and indicate that we do not fundamentally understand if it is related to the differences in enhancement of $PM_{2.5}$ or just a coincidence. Since the difference in spatial variation of $PM_{2.5}$ enhancement and air mass transport is similar between years 2016 and 2015 than between 2018 and 2019, but the concentrations remain similar between 2016 and 2015 but drop from 2018 to 2019, we conclude that the observations at the BUCT-AHL station might be connected to emissions outside the prohibition area with varying extent between different years. The influence of banned firework burning on the sources related to CNY festivities other than fireworks might have an impact on the overall emissions within and outside the ban area, but presumably this impact would be lesser than that of the transportation of air masses and firework plumes.

This discussion is now in lines 462-482 of the revised manuscript:

> Figure 11 shows differences between the $PM_{2.5}$ mean of the sites inside the 5[th] Ring Road and the mean of the sites outside the 5[th] Ring (that is the mean of the 8 inside sites minus the mean of the outside 4 stations) 48 hours before through 48 hours after the CNY for 2013-2019. In 2013, 2014 and 2018, the enhancement of $PM_{2.5}$ during the CNY overnight is greater inside than outside the 5[th] Ring Road. However, in 2015 and 2019, as well as immediately after the CNY midnight in 2016, $PM_{2.5}$ was lower inside than outside. While we were lacking the detailed data on local meteorology during 2013-2016, we were still able to analyze the meteorological condition in terms of air mass trajectories. Figure 4 shows that, similar to 2019 as discussed previously, in 2015 and CNY midnight of 2016, airmasses arriving to Beijing were from the cleaner West-North sector and arrived with much higher velocity in comparison to years 2013, 2014 and 2018, during which the air masses made a turn in South or East before arrival to Beijing. Even though the CNYs during which the increase of PM2.5 enhancement inside the 5[th] Ring Road is less pronounced than outside seem to be related to faster arrival of cleaner airmasses, we have no clear view for the reason of this difference and, due to the qualitative nature of this comparison, it is well possible that this connection is pure coincidence. The similarity of years 2015 and 2019 in terms of the spatial variation of CNY midnight pollution peak suggests that meteorology may be at least part of the reason for the lesser enhancement of pollution levels inside the 5[th] Ring Road than outside. Nevertheless, the notably lower concentrations of $PM_{2.5}$ and gaseous air pollutants in 2019 than in 2015 indicates that, even with similarities in spatial distribution of changes in concentrations, the most likely reason for lower concentrations during CNY night are the lower emissions.

Line 133: The authors introduce the nano-SMPS and NAIS in Section 2.2, respectively. Maybe some discussions about the PSD of nucleation mode measured by the two instruments should be given.

The reviewer is correct that introduction of the instruments in the Methods section was not consistent with the results regarding particle number size distributions. In the revised version, we have modified the analysis of particle number size distributions thoroughly. Instead of showing the results from NAIS, which only covers particle sizes up to 40 nm, we now present the size distributions from the PSD system, covering the size distribution from 1 nm to over 1000 nm (we cut the figures to 1000 nm since the number concentrations in larger sizes are negligible). The observation of particles with diameters of around 20 nanometers appearing simultaneous to the firework emissions turned out to be due to instrumental reasons. Thus, the size ranges most relevant for firework emissions are of diameters larger than 40 nm and using a single instrument (PSD) is more practical than combining NAIS and SMPS or

DMPS (for the CNY 2018 only SMPS data were available, for 2019 only DMPS). Based on this, we have fully rewritten the Section 3.3 (see modified manuscript) and modified the Methods section to reflect this, and we have replaced Figure 3 (now Figure 5) with PSD.

Line 232: The information of meteorological conditions should be more discussed and displayed in figures in section 3.2 so as to more fully explain that the influence of meteorological conditions is relatively small.

We have revised the meteorology section, and included analysis of Hysplit back-trajectories, as mentioned above.

Line 248: Both the 2018 and 2019 CNY are feedbacks after the implementation of the policy. Apart from the effect of fireworks celebration on the level of PM2.5, other sources should also be discussed during the CNY period. During CNY in 2018, the enhancement in PM2.5 inside the 5th Ring is higher than the outside. Is this the influence from other sources?

In terms of other sources, it is very difficult to estimate how the other sources than firework burning would change between the years. As indicated in our response to the referee's comments for manuscript lines 334-346 above, we estimate that the impact of changes in other sources is lesser than that of air mass transport.

Line 320: A long term multi dataset was used to further demonstrate the effect of the policy in reducing PM2.5. But there appear diversity differences between inside and outside 5th Ring from 2013 to 2016. And there is no obvious difference between outside and inside the 5th Ring during 2019 CNY as shown in Fig.8, with one exception.

The differences between the years are discussed in our response to the referee's comments for manuscript lines 334-346 above. Additionally, we want to note the following: Ignoring the one red dot (enhancement of PM2.5 by > factor 4) the other outside data points have enhancement factors between 2.3 and 3. Three out of the eight observation sites inside the $5^{th}$ Ring show similar enhancement factors, but five sites show factors of 2 or smaller. With these differences, the median enhancement becomes larger outside the $5^{th}$ Ring than inside, as seen in Fig. 10. Based on the referee's comment, we added some additional commentary on Figure 8 (now Figure 10) in the manuscript (lines 449-460 in the revised manuscript).

> Figure 10 illustrates that in 2013 and 2014, the enhancement in $PM_{2.5}$ concentrations during CNY is greater inside the $5^{th}$ ring than outside. In 2015, the enhancement is much greater at the two northeastern sites (HR and SY). In 2016, the differences vary, with no clear difference inside or outside the $5^{th}$ ring. In 2018, the enhancement of $PM_{2.5}$ is higher inside the $5^{th}$ ring than outside, except for the SY site to the far northeast, which had significantly high enhancement compared to the other sites. In 2019 the enhancement is overall less inside compared to outside. The enhancement factors outside the $5^{th}$ Ring road (excluding the single highest value) and at the Northern inside stations nearest to the Ring road are quite similar in 2019, roughly in range 2.5 to 3, but the peak times of pollution are few hours earlier at the outside stations than the Northern inside stations (Fig. S12). The measurement sites closer to central Beijing, on the other hand, show clearly lower enhancement factors, of values of two or below. Based on these spatial and temporal differences, and on the northerly winds observed at the time, it is possible that the higher enhancement factors inside but close to the $5^{th}$ Ring road are related to emissions from outside the Ring road.

We have also added further discussion using meteorology and Hysplit to better explain the context and ambient conditions during the 2013-2019 CNY.

Line 341: In 2015 and the first part of 2016, the enhancement of PM2.5 between inside and outside the 5th ring is opposite when compared with most years. How does the author consider the differences in 2015 and 2016 in Figure 9?

These features are discussed in the response to the referee's comments for manuscript lines 334-346 above.

Line 390: The link of the dataset during the CNY in 2018 and 2019 should be given in the acknowledgements.

We have added the following Data Availability statement:

> Data from the BUCT station is available at request. Real time data from the MEP stations is available at http://106.37.208.233:20035/. Archived, quality-controlled MEP data may also be available upon request.

Minor Comments

Some abbreviations are incorrect, e.g. Line 112: "PM2.5","PM10", please check the whole text.

We thank the reviewer for this suggestion. We have gone through the manuscript and have properly subscripted these, as well as chemical compounds, consistently throughout the manuscript.

Line 52: add the definition before "CNY", because this is the first time it appears in the introduction.

We have added this definition of the acronym.

Section 2.2.1: The authors should give more detailed descriptions about the instruments.

We have added a table in supplementary material, Table S2, with details of the instruments used in this study. We have added the following text (lines 138-140 in the revised manuscript):

> Technical details of the instruments, including manufacturer, parameters measured, time resolution, and available time periods of measurements, can be found in Table S2 in Supplementary material. These details are also described in Liu et al. (2020).

Line 165: should be "Kürten et al. (2012)".

We have fixed this. Thank you for pointing this out to us.

Line 176: should be "Song et al., 2017; Tao et al., 2016", please check the citation format throughout the paper.

We have fixed this formatting throughout the manuscript, specifically fixed the lack of commas after et al. Thank you for pointing this out.

Some descriptions in this paper should be more quantitative, such as line 303: "there was a spike in pollution around midnight during the CNY".

We have added some quantitative values to this paragraph. It now reads as follows (lines 401-409 in the revised manuscript):

Figure 8 shows that each year, there was a spike in pollution around midnight during the CNY. The highest levels were observed in 2016, where the peak in $PM_{2.5}$ around midnight of the CNY reaching almost 700 µg/cm$^3$ while values earlier in the day were less than 100 µg/cm$^3$. The lowest levels of $PM_{2.5}$ were in 2019 with the overnight peak less than µg/cm$^3$ compared to daytime values around 50 µg/cm$^3$. Observations from 2013, 2014, 2015, and 2017 also showed similarly high or higher levels of $PM_{2.5}$ as in 2018 (unfortunately the 2017 dataset is incomplete and does not extend beyond 00:00 of New Years day). These measurements for all seven years are in agreement with other studies that have linked elevated air pollution levels to CNY celebrations (Yang et al., 2014; Shi et al., 2014; Feng et al., 2012; Zhang et al., 2010), and this study further shows that the peak in 2019 is lower than in 2018, which is lower than in 2016 and 2017.

Lines: 306-308: The author attribute the pollution events following CNY to firework burning as well, please give more evidences.

In the introduction, we have cited 10 studies linking a spike in pollution to holiday celebrations involving fireworks, including CNY and other similar holidays around the world. The results of this study are in agreement with these studies, and furthermore this study shows that the peak in 2019 is less than in 2016. To help clarify this point, we have added quantifications to the statement, and we also added citations at the end of this paragraph. A new sentence was added to the end of the paragraph (lines 406-409 in the revised version):

> The measurements for all seven years are in agreement with other studies that have linked elevated air pollution levels to CNY celebrations (Yang et al., 2014; Shi et al., 2014; Feng et al., 2012; Zhang et al., 2010), and this study further shows that the peak in 2019 is lower than in 2018, which is lower than in 2016 and 2017.

7: The pollutant concentrations seem low in 2013 and 2014 compared to other years before prohibition on firework, do you attempt to figure out the reason?

Thank you for pointing out this discrepancy in our result.

Going through all the data meticulously, we spotted an unfortunate error in the treatment of NaN values in the script applied for making this figure. The corrected figure is Figure 9 in the current version of the manuscript.

Section 3.5: You should compare your results with previous studies. It is best to add more references in these paragraphs.

There are no references here because we did not find any previous studies that studies spatial variability of holiday related pollution that investigated multiple sites across a large metropolitan area. This is one of the novelties of this particular study. Nonetheless, based on this comment and other referee comments, we have made a few revisions to this section and to the conclusions to make this point clearer.

**Other**

We realized there was a mistake in Figure 2 (a), for $PM_{2.5}$. In this timeseries figure, the values for 2019 were off by one day, with the line shown one day later than it should have been. The revised manuscript now contains the corrected figure.

We fixed Figure 6, which now shows the correct values for aerosol particle concentrations.

We also corrected a mistake in the boxplots, which was miscalculating NaN values.

The name of co-author Kaspar R. Daellenbach was misspelled, which has been corrected.
* * *
We once again thank the reviewers and editors for their time and effort put into helping us revise and improve this manuscript.

Sincerely,
Benjamin Foreback
Institute for atmospheric and Earth system Research (INAR)/Physics
University of Helsinki

---

## Referee Report (RR1)

This manuscript has improved after the revisions. The authors have solved most of the pre-existing problems. But there are still some unanswered questions. I recommend this paper to publish after the authors deals with the below problems.

1. The language in the revised manuscript needs further improvement. Some low-level grammatical errors are easily found, such as the errors in line 46, 82, 182, 334, 487. I recommend to find a native English speaker to help correct this paper before next submission.
2. The authors stated that the study is the first using the CNY data covering several years. Actually, some previous studies have used more data to study this issue, such as Sun et al. (2020). The authors need to refer more papers in the introduction part.

**Sun, Y., Lei, L., Zhou, W., Chen, C., He, Y., Sun, J., ... & Worsnop, D. R. (2020). A chemical cocktail during the COVID-19 outbreak in Beijing, China: Insights from six-year aerosol particle composition measurements during the Chinese New Year holiday. Science of the Total Environment, 742, 140739.**

3. In section 3.1, the authors attributed high concentrations of $SO_2$ and BC on CNY in 2018 to fireworks. However, during the 2018 CNY period fireworks were prohibited. The coal combustion can also emit a lot of $SO_2$ and BC. Previous studies (e.g., Wang et al. 2018) suggested that massive emission of $SO_2$ from coal-based power plants, steel and iron works, glassworks and cement mills in the southern Hebei province, the south of Beijing. Therefore, the high-level air pollutants on CNY in 2018 may be from long distance transportation.

**Wang, Y., Li, Z., Zhang, Y., Du, W., Zhang, F., Tan, H., Xu, H., Fan, T., Jin, X., Fan, X., Dong, Z., Wang, Q., and Sun, Y.: Characterization of aerosol hygroscopicity, mixing state, and CCN activity at a suburban site in the central North China Plain, Atmos. Chem. Phys., 18, 11739-11752, 10.5194/acp-18-11739-2018, 2018.**

4. Line 317-319. The authors said that the air masses transport conditions on CNY in 2018 and 2019 were rather similar. However, the trajectories in Fig.4 show that air mass transported from the south to Beijing in CNY 2018 but from the east to Beijing in CNY 2019. The authors need to know the pollution levels were distinct in the south and east regions of Beijing. Weaker diffusion conditions and more industrial emissions make that the pollution in the south of Beijing is much stronger than in the east (e.g., Wang et al. 2019). Therefore, the transportation from industrial pollutants should be stronger in CNY 2018.

**Wang, Y., Dörner, S., Donner, S., Böhnke, S., De Smedt, I., Dickerson, R. R., Dong, Z., He, H., Li, Z., Li, Z., Li, D., Liu, D., Ren, X., Theys, N., Wang, Y., Wang, Y., Wang, Z., Xu, H., Xu, J., and Wagner, T.: Vertical profiles of NO2, SO2, HONO, HCHO, CHOCHO and aerosols derived from MAX-DOAS measurements at a rural site in the central western North China Plain and their relation to emission sources and effects of regional**

transport, Atmos. Chem. Phys., 19, 5417-5449, 10.5194/acp-19-5417-2019, 2019.

5. Figure 2i: Why the BLH was larger than 2 km in so much time, especially in 2018? In general, BLH was lower than 1 km in winter.
6. Figure 10 was too vague. The full names of these measurement sites need to be annotated.

---

## Author Response (AR2)

Response to Referee and Editor on "Measurement Report: A Multi-Year Study on the Impacts of Chinese New Year Celebrations on Air Quality in Beijing, China."

22 February 2022

Dear Editor and Referee:

Thank you for your time in re-reviewing our revised manuscript. We appreciate the helpful comments to make further improvements to our manuscript prior to publication. We have addressed the comments and have made minor modifications to the manuscript. Our replies to the comments are in blue below.

**Editor Comments**

Please note that your reference list has not been compiled according to our standards. Please consider adjusting your reference list with the next revision of your manuscript. The manuscript preparation guidelines can be seen at: https://www.atmospheric-chemistry-and-physics.net/for_authors/manuscript_preparation.html.

> We have updated the format of references to reflect that of Copernicus publications.

1. Regarding your figure 10: With the next revision, please add the copyright symbol as follows: © Google Earth (either in the figure itself or in the caption). Please also add the corresponding author below the affiliations.

> We have added the copyright statement to the figure caption. We have also noted the corresponding author.

2. The section "Correspondence to:" is missing from your *.pdf manuscript file. Please indicate it for the next revision.

> We have added "Correspondence to: Pauli Paasonen (pauli.paasonen@helsinki.fi)" after the author list and affiliations.

**Referee Comments**

1. The language in the revised manuscript needs further improvement. Some low-level grammatical errors are easily found, such as the errors in line 46, 82, 182, 334, 487. I recommend to find a native English speaker to help correct this paper before next submission.

> Thank you for pointing out these typos in the text. We have corrected them.

2. The authors stated that the study is the first using the CNY data covering several years. Actually, some previous studies have used more data to study this issue, such as Sun et al. (2020). The authors need to refer more papers in the introduction part.

Sun, Y., Lei, L., Zhou, W., Chen, C., He, Y., Sun, J., ... & Worsnop, D. R. (2020). A chemical cocktail during the COVID-19 outbreak in Beijing, China: Insights from six-year aerosol particle composition measurements during the Chinese New Year holiday. Science of the Total Environment, 742, 140739.

Thank you for pointing out that this statement was not entirely clear, and as a result, it could be misinterpreted. We would like to clarify that we haven't claimed that this is the first study, rather we stated (line 111), "it is one of the only studies to not only show measurements for a single CNY…". To address the referee's comment and make the statement clearer, we have rephrased this as, "it is one of only a few studies to not only show measurements for a single CNY…"

3. In section 3.1, the authors attributed high concentrations of SO2 and BC on CNY in 2018 to fireworks. However, during the 2018 CNY period fireworks were prohibited. The coal combustion can also emit a lot of SO2 and BC. Previous studies (e.g., Wang et al. 2018) suggested that massive emission of SO2 from coal-based power plants, steel and iron works, glassworks and cement mills in the southern Hebei province, the south of Beijing. Therefore, the high-level air pollutants on CNY in 2018 may be from long distance transportation.

Wang, Y., Li, Z., Zhang, Y., Du, W., Zhang, F., Tan, H., Xu, H., Fan, T., Jin, X., Fan, X., Dong, Z., Wang, Q., and Sun, Y.: Characterization of aerosol hygroscopicity, mixing state, and CCN activity at a suburban site in the central North China Plain, Atmos. Chem. Phys., 18, 11739-11752, 10.5194/acp-18-11739-2018, 2018.

Thank you for the description of emissions from fireworks versus other combustion sources. Although it is true that the pollutants could have been transported from southern Hebei province, this does clearly not explain the sudden increase followed by sudden decline that we see in Figures 2, 5, 8 and 11. We took note of the trajectory plots (Figure 4): Trajectories from 6 hours before CNY up to CNY are from southern/southwestern Hebei province, which means that if the high concentrations of observed $SO_2$ and BC were due to transport from this region, the high concentrations would have been seen during the whole overnight period, rather than a sudden peak around midnight. We can therefore infer that the observation is due to a sudden short-term emission source rather than long-term transport. Comparing this to our spatial analysis in Figures 10 and S2-S13, we see that the peaks occur in all parts of the metropolitan area at roughly the same time. This indicates that these short-term emissions are likely sourced within the metropolitan area because if the spike was transported from the south, there would be a delay between the observations in the south part of the city and the north part of the city.

We have made a minor revision to the paragraph starting at line 255 in the previous revision. In the new revision (now lines 268-276), this paragraph reads as follows:

The measurements showed elevated nighttime concentration of $H_2SO_4$ on CNY in 2018 exceeding $3 \cdot 10^6$ cm$^{-3}$ during the whole night, which was an order of magnitude higher than typical nighttime $H_2SO_4$ concentrations of $5 \cdot 10^5$ cm$^{-3}$ (Dada et al., 2020). In 2019, there was no evident indication of anomalies in nighttime $H_2SO_4$ concentration during CNY. An unknown spike in $H_2SO_4$ was noticed at noon the day before CNY in 2018, and its association with celebratory activities is unclear. Like with $PM_{2.5}$ and $SO_2$, Figure 2 shows a distinctive spike in BC around midnight of the 2018 CNY. Although $SO_2$ and BC also originate from coal combustion and other emission sources (Wang et al., 2018), because of the shortness of the peak, and the fact that it occurs at exactly midnight, these simultaneous peaks of BC and SO2 during the nighttime of CNY most likely originate from firework burning.

We have added Wang et al., 2018 to the reference list.

4. Line 317-319. The authors said that the air masses transport conditions on CNY in 2018 and 2019 were rather similar. However, the trajectories in Fig.4 show that air mass transported from the south to Beijing in CNY 2018 but from the east to Beijing in CNY 2019. The authors need to know the pollution levels were distinct in the south and east regions of Beijing. Weaker diffusion conditions and more industrial emissions make that the pollution in the south of Beijing is much stronger than in the east (e.g., Wang et al. 2019). Therefore, the transportation from industrial pollutants should be stronger in CNY 2018.

Wang, Y., Dörner, S., Donner, S., Böhnke, S., De Smedt, I., Dickerson, R. R., Dong, Z., He, H., Li, Z., Li, Z., Li, D., Liu, D., Ren, X., Theys, N., Wang, Y., Wang, Y., Wang, Z., Xu, H., Xu, J., and Wagner, T.: Vertical profiles of NO2, SO2, HONO, HCHO, CHOCHO and aerosols derived from MAX-DOAS measurements at a rural site in the central western North China Plain and their relation to emission sources and effects of regional transport, Atmos. Chem. Phys., 19, 5417-5449, 10.5194/acp-19-5417-2019, 2019.

Thank you for the detailed explanation of sources from the south as opposed to the east. Thank you, also, for pointing out that we imply they are the same in 2018 and 2019, even though they are not. The referee is correct that in 2018, the airmass 6 hours through 2 hours prior to CNY midnight comes from southwest before arriving at the station, whereas in 2019, the airmass from 6 hours to 2 hours prior to CNY midnight arrive from the east.

The referee is also correct that the emissions from southwest and east are not the same, which we incorrectly stated were "rather similar." The referee is further correct that we would expect air from the southwest to be more polluted than air to the east. However, what we see is different. Based on the Wang et al. (2019) paper, the air from 6 hours before CNY through 2 hours before CNY in 2019 (where airmass is from the east) should be cleaner than the air from 6 hours through 2 hours before CNY in 2018 (which is from the southwest). What is observed, on the other hand, shown in Figure 2, is that during this time frame – before the sudden peak is observed, which we could call the "background value" in this case – PM$_{2.5}$ concentrations in 2019 are higher than in 2018.

Nonetheless, the sudden peak around midnight with respect to the background concentrations is indicative of a short-term, probably nearby, emission source as opposed to long-range transport. From this, we can conclude that in both years, the sharp, short-term peak is from the short-term source, i.e. the firework burning.

To address the referee's comment regarding our incorrect statement that the trajectories are rather similar, we have corrected the language in this paragraph to explain more clearly and correctly. We have removed the misleading statement that the trajectories are "rather similar." The new language is now consistent with the figure.

This revised paragraph (lines 334-349 in the revised manuscript), which is now split into three paragraphs, reads as follows:

The lower concentrations observed during the emission spike in 2019 can be either due to lower emission rates in the area with which the measured air mass is in contact, or due to a shorter exposure to roughly similar emissions during both years. Figure 4 shows 96 hour back-trajectories by Hysplit, during the night of CNY in 2018 and 2019, showing the sources of the airmasses. This provides further insights into the history of the

airmasses in Beijing, including how clean we can expect the airmasses to be before CNY, and whether the airmasses are stagnant around Beijing or whether clean air is being transported into the city.

These trajectories show the following: In 2018, the airmasses from six hours prior to CNY through CNY are from the southwest, and from two through six hours after CNY, the airmass is from the west. In 2019, airmasses from six hours prior to CNY through two hours prior to CNY the airmasses are from the east, and following the CNY the airmasses are primarily from the west.

Based on Wang et al. (2019), airmasses from the east are expected to be cleaner than from the southwest due to more diffusion and less emissions from industry. However, we observed the opposite: From six through two hours prior to midnight (i.e. the background value before the spike in pollution), the background pollutant concentrations are higher in 2019 than in 2018. This gives further indication that the emission sources are likely localized and short-term as opposed to long-range transport.

Wang et al. (2019) has been added to the reference list.

5. Figure 2i: Why the BLH was larger than 2 km in so much time, especially in 2018? In general, BLH was lower than 1 km in winter.

We have done an investigation and have verified that these measurements are correct. During the daytime, the BLH is greater than 2 km during times of clear or relatively clean air, which is in general the case during the daytime with clean conditions in this time period. Winter is the most polluted season in Beijing and therefore many studies are focusing on the highly polluted periods with low BLH. During periods of haze/air pollution episodes, BLH may remain less than 2 km during the daytime because of the radiative effects of aerosols, which is attenuating the incoming solar radiation, which decreases the energy reaching the surface and increases the atmospheric stability within the boundary layer, consequently keeping the BLH low. Since this is not the case at all times during the daytime in this time period, it is reasonable to have BLH >2km during the cleaner days.

Here are a couple of studies showing that 2 km is a reasonable value for daytime BLH in Beijing during times of clear air:

Liu, Q., Jia, X., Quan, J. *et al.* New positive feedback mechanism between boundary layer meteorology and secondary aerosol formation during severe haze events. *Sci Rep* **8,** 6095 (2018). https://doi.org/10.1038/s41598-018-24366-3

Jiang, Q.; Zhang, H.; Wang, F.; Wang, F. Research on the Growth Mechanism of $PM_{2.5}$ in Central and Eastern China during Autumn and Winter from 2013–2020. *Atmosphere* **2022**, *13*, 134. https://doi.org/10.3390/atmos13010134

6. Figure 10 was too vague. The full names of these measurement sites need to be annotated.

Thank you for pointing out that the figure could use some improvement for clarity.

The figure contains the abbreviations of each station next to the station. However, we realize that in grey font, it is somewhat difficult to see against the imagery. We have modified the annotations to be slightly brighter. Additionally, a table of the twelve MEP sites with their names in Chinese, abbreviations in English, and their latitude and longitude values is in the Supplementary material.

We have now added the site names translated to English in addition to the abbreviations to this table. We have added to the figure caption a reference to the table.

The figure and its caption are now as follows:

[Figure]

**Figure 10:** The 12 MEP sites mapped in the Beijing metropolitan area, showing the ratio of overnight $PM_{2.5}$ observations during the CNY (21:00-05:00) to all data during the period of 48 hours before through 48 hours after the CNY. The red line marks the approximate location of the 5[th] Ring Road. Note that the colorbars in each map are relative to only that year, and the colorbar range is not the same in different years. 2017 is omitted from this figure because data after 00:00 was not available. A list of the sites' full names in English and Chinese, along with their latitude and longitude coordinates can be found in Table S1 in Supplementary material. Imagery: © Google Earth.

Table S1 has been revised as follows:

| Site name | Site name (eng) | Site abbreviation (eng) | Longitude | Latitude |
|---|---|---|---|---|
| 东四 | DongSi | DS | 116.417 | 39.929 |
| 天坛 | TianTan | TT | 116.407 | 39.886 |
| 官园 | GuanYuan | GY | 116.339 | 39.929 |
| 万寿西宫 | WanShouXiGong | WSXG | 116.352 | 39.878 |
| 奥体中心 | AoTiZhongXin | ATZX | 116.397 | 39.982 |
| 农展馆 | NongZhanGuan | NZG | 116.461 | 39.937 |
| 万柳 | WanLiu | WL | 116.287 | 39.987 |
| 古城 | GuCheng | GC | 116.184 | 39.914 |
| 顺义 | ShunYi | SY | 116.655 | 40.127 |
| 昌平 | ChangPing | CP | 116.23 | 40.217 |
| 怀柔 | HuaiRou | HR | 116.628 | 40.328 |
| 定陵 | DingLing | DL | 116.22 | 40.292 |

**Table S1:** List of the 12 MEP (Ministry of Environmental Protection) sites in the Beijing metropolitan area, their translations and abbreviations in English, and their geographic locations.

**Other:**

The affiliation for one of the co-authors, Rosaria E. Pileci, has changed. The new version of this manuscript reflects her new affiliation.

We have moved the figures to be integrated into the text instead of appended at the end, per guidelines: "Figures and tables as well as their captions must be inserted in the main text near the location of the first mention (not appended to the end of the manuscript)."

When uploading our final revision, we will also upload the figures separately in pdf format.